# Dynamics of a colloidal particle coupled to a Gaussian field: From a confinement-dependent to a non-linear memory

**Urna Basu[1,2], Vincent Démery[3,4] and Andrea Gambassi[5*]**

**1** S. N. Bose National Centre for Basic Sciences, JD Block, Saltlake, Kolkata 700106, India
**2** Raman Research Institute, C. V. Raman Avenue, Bengaluru 560080, India
**3** Gulliver UMR CNRS 7083, ESPCI Paris, Université PSL,
10 rue Vauquelin, 75005 Paris, France
**4** ENSL, CNRS, Laboratoire de physique, F-69342 Lyon, France
**5** SISSA — International School for Advanced Studies and INFN,
via Bonomea 265, 34136 Trieste, Italy

⋆ gambassi@sissa.it

## Abstract

The effective dynamics of a colloidal particle immersed in a complex medium is often described in terms of an overdamped linear Langevin equation for its velocity with a memory kernel which determines the effective (time-dependent) friction and the correlations of fluctuations. Recently, it has been shown in experiments and numerical simulations that this memory may depend on the possible optical confinement the particle is subject to, suggesting that this description does not capture faithfully the actual dynamics of the colloid, even at equilibrium. Here, we propose a different approach in which we model the medium as a Gaussian field linearly coupled to the colloid. The resulting effective evolution equation of the colloidal particle features a non-linear memory term which extends previous models and which explains qualitatively the experimental and numerical evidence in the presence of confinement. This non-linear term is related to the correlations of the effective noise via a novel fluctuation-dissipation relation which we derive.



# 1   Introduction

Since the very early days of statistical physics, the motion of a Brownian particle in a simple fluid solvent has been successfully described by means of a linear Langevin equation for its velocity $v$ [1]. In particular, the interaction of the mesoscopic particle with the microscopic molecular constituents of the solvent generates a deterministic friction force — often assumed to have the linear form $-\gamma v$ — and a stochastic noise, the correlations of which are related to the friction by a fluctuation-dissipation relation. This relationship encodes the equilibrium nature of the thermal bath provided by the fluid. When the solvent is more complex or the phenomenon is studied with a higher temporal resolution, it turns out that the response of the medium to the velocity of the particle is no longer instantaneous, for instance due to the hydrodynamic memory [2] or to the possible viscoelasticity of the fluid [3]. In this case, the solvent is characterised by a retarded response $\Gamma(t)$ that determines the effective friction $-\int^t dt' \Gamma(t-t')v(t')$. Experimentally, the memory kernel $\Gamma(t)$ can be inferred from the spectrum of the equilibrium fluctuations of the probe particle. Microrheology then uses the relation between the memory kernel $\Gamma(t)$ and the viscoelastic shear modulus of the medium to infer the latter from the observation of the motion of the particle [4]. In recent years, this approach has been widely used for probing soft matter and its statistical properties [5–7]. Accordingly,

it is important to understand how the properties of the medium actually affect the static and dynamic behaviour of the probe.

Although the description of the dynamics outlined above proved to be viable and useful, it has been recently shown via molecular dynamics simulations of a methane molecule in water confined by a harmonic trap that the resulting memory kernel turns out to depend on the details of the confinement [8]. A similar effect has been also observed for a colloidal particle immersed in a micellar solution [9]. This means that the memory kernel $\Gamma(t)$ does no longer characterise the very interaction between the particle and the medium, as it depends significantly on the presence of additional external forces. Here we show, on a relatively simple but representative case and within a controlled approximation, that this dependence — theoretically expected but practically often forgotten — is actually due to the fact that the effective dynamics of the probe particle is ruled by a *non-linear* evolution equation.

In order to rationalise their findings, the experimental data for a colloidal particle immersed in a micellar solution were compared in Ref. [9] with the predictions of a stochastic Prandtl-Tomlinson model [10, 11], in which the solvent, acting as the environment, is modelled by a fictitious particle attached to the actual colloid via a spring. This simple model thus contains two parameters: the friction coefficient of the fictitious particle and the stiffness of the attached spring. While this model indeed predicts a confinement-dependent friction coefficient for the colloid, the fit to the experimental data leads anyhow to confinement-dependent parameters of the environment. Accordingly, as the original modelling, it does not actually yield a well-characterised bath-particle interaction, which should be independent of the action of possible additional external forces.

Although the equilibrium and dynamical properties of actual solvents can be quite complex, here we consider a simple model in which the environment consists of a background solvent and a fluctuating Gaussian field $\phi$ with a relaxational and locally conserved dynamics and with a tunable correlation length $\xi$, also known as model B [12]. The motion of the colloid is then described by an overdamped Langevin equation, with an instantaneous friction coefficient determined by the background solvent and a linear coupling to the Gaussian field [13, 14]. The colloid and the field are both in contact with the background solvent, acting as a thermal bath. In practice, the possibility to tune the correlation length $\xi$ (and, correspondingly, the relaxation time of the relevant fluctuations) is offered by fluid solvents thermodynamically close to critical points, such as binary liquid mixtures, in which $\xi$ diverges upon approaching the point of their phase diagrams corresponding to the demixing transition [12]. This model and variations thereof have been used in the literature in order to investigate theoretically the dynamics of freely diffusing or dragged particles [13, 15, 16], in the bulk or under spatial confinement [14] as well as the field-mediated interactions among particles and their phase behavior [17–19]. Here, instead, we use it in order to rationalise in a simple and natural setting the possible emergence of the issues mentioned above in interpreting microrheology data in the presence of correlated media.

The simplified model discussed above allows us to address another relevant and related question, i.e., how the behaviour of a tracer particle is affected upon approaching a phase transition of the surrounding medium. Among the aspects which have been investigated theoretically or experimentally to a certain extent we mention, e.g., the drag and diffusion coefficient of a colloid in a near-critical binary liquid mixture [20–23] or the dynamics and spatial distribution of a tracer particle under spatial confinement [14]. However, the effective dynamical behaviour and fluctuations of a tracer particle in contact with a medium near its bulk critical point is an issue which has not been thoroughly addressed in the literature. The model we consider here allows us to explore at least some aspects of this question, with the limitation that we keep only the conservation of the order parameter $\phi$ as the distinguishing feature of the more complex actual dynamics of a critical fluid [12].

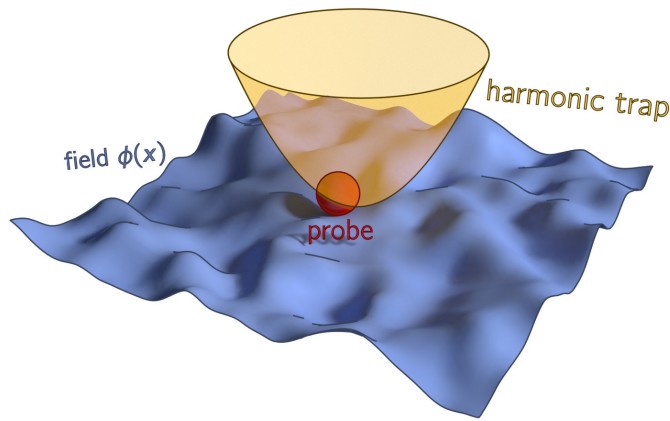

Figure 1: Schematic representation of the model: The colloidal probe particle (dark red sphere) is in contact with a fluctuating scalar field $\phi(x)$ and is subject to the force exerted by a harmonic trap.

Similarly, in some respect, the model considered here provides an extension of the stochastic Prandtl-Tomlinson model in which the colloid has a non-linear interaction with a large number of otherwise non-interacting fluctuation modes of the field, which play the role of a collection of fictitious particles.

We show that an effective dynamics can be written for the probe position $X$, which features a time- and $X$-dependent memory, thereby rendering the resulting dynamics non-linear and non-Markovian. In particular, as sketched in Fig. 1, we consider a particle which is also confined in space by a harmonic trap generated, for instance, by an optical tweezer. We then compute, perturbatively in the particle-field coupling, the two-time correlation function $C(t) = \langle X(0)X(t)\rangle$ in the stationary state. From $C(t)$ we derive the associated effective *linear* memory kernel $\Gamma(t)$. This effective memory turns out to depend on the strength of the confinement qualitatively in the same way as it was reported in the molecular dynamics simulations and in the experiments mentioned above and this dependence is enhanced upon approaching the critical point, i.e., upon increasing the spatio-temporal extent of the correlations within the medium.

In addition, we show that the correlation function $C(t)$ generically features an algebraic decay at long times, with specific exponents depending on the Gaussian field being poised at its critical point or not. In particular, we observe that generically, in $d$ spatial dimensions, $C(t) \propto t^{-(1+d/2)}$ at long times $t$, due to the coupling to the field, while the decay becomes even slower, with $C(t) \propto t^{-d/4}$, when the medium is critical. Correspondingly, the power spectral density $S(\omega)$, which is usually studied experimentally, displays a non-monotonic correction as a function of $\omega$ which is amplified upon approaching the critical point and which results, in spatial dimensions $d < 4$, in a leading algebraic singularity of $S(\omega) \propto \omega^{-1+d/4}$ for $\omega \to 0$ at criticality.

The results of numerical simulations are presented in order to demonstrate that the qualitative aspects of our analytical predictions based on a perturbative expansion in the particle-field interaction actually hold beyond this perturbation theory.

The rest of the presentation is organised as follows: In Sec. 2 we define the model, discuss its static equilibrium properties and the correlations of the particle position in the absence of the coupling to the field. In Sec. 3 we discuss the non-linear effective dynamical equation for the motion of the particle, we introduce the linear memory kernel, we derive the generalised fluctuation-dissipation relation which relates the non-linear friction to the correlations of the non-Markovian noise and we work out its consequences. In Sec. 4 we proceed to a perturba-

tive analysis of the model, expanding the evolution equations at the lowest non-trivial order in the coupling parameter between the particle and the field. In particular, we determine the equilibrium correlation function $C(t)$ of the particle position, discuss its long-time behaviour depending on the field being poised at criticality or not. Then we investigate the consequences on the spectrum of the dynamic fluctuations, inferring the effective memory kernel and showing that it actually depends on the parameters of the confining potential. We also investigate the relevant limits of strong and weak trapping. The predictions derived in Sec. 4, based on a perturbative expansion, are then confirmed in Sec. 5 via numerical simulations of a random walker interacting with a one-dimensional chain of Rouse polymer. Our conclusions and outlook are presented in Sec. 6, while a number of details of our analysis are reported in the Appendices.

## 2 The Model

### 2.1 Probe particle coupled to a Gaussian field

We consider a "colloidal" particle in contact with a fluctuating Gaussian field $\phi(x)$ in $d$ spatial dimensions. The particle is additionally trapped in a harmonic potential of strength $\kappa$, centered at the origin of the coordinate system. The total effective Hamiltonian $\mathcal{H}$ describing the combined system is

$$\mathcal{H}[\phi(x),X] = \int d^d x \ \left\{ \frac{1}{2}[\nabla\phi(x)]^2 + \frac{r}{2}\phi^2(x) \right\} + \frac{\kappa}{2}X^2 - \lambda \int d^d x \ \phi(x)V(x-X), \quad (1)$$

where the vector $X = \{X_1, X_2, \cdots, X_d\}$ denotes the position of the colloid, $V(z)$ is the coupling between the medium and the particle, and $r > 0$ controls the spatial correlation length $\xi = r^{-1/2}$ of the fluctuations of the field. For convenience, we introduce also a dimensionless coupling strength $\lambda$ which will be useful for ordering the perturbative expansion discussed in Sec. 4. The field $\phi(x)$ undergoes a second-order phase transition at the critical point $r = 0$ and its spatial correlation length $\xi$ diverges upon approaching it. The interaction between the particle and the medium is physically expected to occur primarily within the space occupied by the particle and therefore $V(z)$ in Eq. (1) will be concentrated in a neighbourhood of $z = 0$ of linear extension corresponding to its radius $R$. For example, in a concrete case, we will assume that $V(z)$ has a Gaussian dependence on $|z|$ (Eq. (52)). Most of the expressions derived below apply to a generic choice of a $V(z)$ which is isotropic in space, i.e., a function of $|z|$, having in mind spherical particles. However, the generalisation to anisotropic interactions (i.e., to anisotropic particles) is generally straightforward. The choice of a coupling between the particle and the medium which is linear in $\phi$ is motivated by two facts: (i) a colloidal particle inserted in a near-critical medium generically favours one of the two competing phases of the system, i.e., values of the order parameter field $\phi(x,t)$ with a certain sign, therefore breaking the symmetry $\phi(x,t) \leftrightarrow -\phi(x,t)$ of the unperturbed fluctuations; (ii) A linear coupling allows a non-perturbative solution of the dynamics of the field in terms of that of the particle coordinate $X(t)$, which therefore leads to an exact effective dynamics for $X(t)$, as discussed further below.

The colloidal particle is assumed to move according to an overdamped Langevin dynamics and we consider the field to represent a conserved medium so that its dynamics follows the so-called model B of Ref. [12]. The dynamics of the joint colloid-field system then reads [24],

$$\partial_t \phi(x,t) = D\nabla^2 \frac{\delta\mathcal{H}[\phi(x,t),X(t)]}{\delta\phi(x,t)} + \eta(x,t), \quad (2)$$

$$\gamma_0 \dot{X}(t) = -\nabla_X \mathcal{H}[\phi(x,t),X(t)] + \zeta(t). \quad (3)$$

Here $\nabla$ and $\nabla_X$ indicate the derivatives with respect to the field coordinate $x$ and the probe position $X$, respectively. $D$ denotes the mobility of the medium and $\gamma_0$ is the drag coefficient of the colloid; these quantities set the relative time scale between the fluctuations of the field and the motion of the colloid. The noises $\eta(x,t)$ and $\zeta(t)$ are Gaussian, delta-correlated, and they satisfy the fluctuation-dissipation relation [25], i.e.,

$$\langle \eta_i(x,t)\eta_j(x',t')\rangle = -2DT\delta_{ij}\nabla^2\delta^d(x-x')\delta(t-t'), \tag{4}$$

$$\langle \zeta_i(t)\zeta_j(t')\rangle = 2\gamma_0 T\delta_{ij}\delta(t-t'), \tag{5}$$

where $T$ is the thermal energy of the bath. Here we assume that the bath acts on both the particle and the field, such that they have the same temperature.

For the specific choice of the Hamiltonian in Eq. (1), the equations of motion (2) and (3) take the form

$$\partial_t\phi = D\nabla^2[(r-\nabla^2)\phi - \lambda V_X] + \eta, \tag{6}$$

$$\gamma_0\dot{X} = -\kappa X + \lambda f + \zeta, \tag{7}$$

where, for notational brevity, we have suppressed the arguments and denoted $V_X(x) = V(x-X)$. The force $f$ exerted on the colloid by the field through their coupling depends on both the colloid position $X(t)$ and the field configuration $\phi(x,t)$ and it is given by

$$f(X(t),\phi(x,t)) \equiv \nabla_X \int d^d x\, \phi(x,t)V(x-X(t)). \tag{8}$$

We note here, for future convenience, that the equations of motion above are invariant under the transformation

$$\{\phi,\lambda\} \to \{-\phi,-\lambda\}. \tag{9}$$

In this work we will focus primarily on the dynamics of the position $X(t)$ of the colloidal particle and in particular on the equilibrium two-time correlation

$$C(t) = \langle X(t)\cdot X(0)\rangle = d\langle X_j(t)X_j(0)\rangle, \tag{10}$$

for any $j \in \{1,2,\dots d\}$, where in the last equality we have used the rotational invariance which follows by assuming the same invariance for $V(x)$. Below we discuss first the equilibrium distribution and then the correlation function $C(t)$ in the absence of the coupling to the field.

## 2.2 Equilibrium distribution

The equilibrium measure of the joint system of the colloid and the field is given by the Gibbs-Boltzmann distribution:

$$P[X,\phi(x)] \propto \exp\left(-\frac{\mathcal{H}[\phi(x),X]}{T}\right). \tag{11}$$

In equilibrium, the position $X$ of the colloid fluctuates around the minimum of the harmonic trap and the corresponding probability distribution can be obtained as the marginal of the Gibbs measure in Eq. (11). This marginal distribution $P(X)$ is actually independent of $\lambda$ because the coupling between the colloid and the fluctuating medium is translationally invariant in space (see Appendix A) and therefore

$$P(X) \propto \exp\left(-\frac{\kappa X^2}{2T}\right), \tag{12}$$

i.e., $P(X)$ is solely determined by the external harmonic trapping.

## 2.3 Correlations in the absence of the field

In Sec. 4 we will express the correlation function $C(t)$ defined in Eq. (10) in a perturbative series in $\lambda$. The zeroth order term $C^{(0)}(t)$ of this expansion is the correlation of the particle in the absence of coupling to the field, i.e., for $\lambda = 0$. In this case, the position $X(t)$ of the colloid follows an Ornstein-Uhlenbeck process and its two-time correlation is (see, e.g., Ref. [26], Sec. 7.5, and also Sec. 4.1 below)

$$C^{(0)}(t) = \frac{d\,T}{\kappa} e^{-\omega_0 |t|}, \tag{13}$$

where

$$\omega_0 = \kappa/\gamma_0 \tag{14}$$

is the relaxation rate of the probe particle in the harmonic trap. Equation (13) shows also that, as expected, one of the relevant length scales of the system is $\ell_T \equiv \sqrt{T/\kappa}$, which corresponds to the spatial extent of the typical fluctuations of the position of the center of the colloid in the harmonic trap, due to the coupling to the bath.

The power spectral density (PSD) $S^{(0)}(\omega)$ of the fluctuating position of the particle in equilibrium, i.e., the Fourier transform of $C^{(0)}(t)$ in Eq. (13), takes the standard Lorenzian form

$$S^{(0)}(\omega) = \frac{d\,T}{\kappa} \frac{2\omega_0}{\omega_0^2 + \omega^2}. \tag{15}$$

Our goal is to determine the corrections to the two-time correlations $C^{(0)}(t)$ and therefore to $S^{(0)}(\omega)$ due to the coupling $\lambda$ to the field, given that one-time quantities in equilibrium are actually independent of it, as discussed above.

# 3 Effective non-linear dynamics of the probe particle

## 3.1 Effective dynamics

The dynamics of the field $\phi$ in Eq. (6) is linear and therefore it can be integrated and substituted in the evolution equation for $X$ in Eq. (7), leading to an effective non-Markovian overdamped dynamics of the probe particle [13,15]

$$\gamma_0 \dot{X}(t) = -\kappa X(t) + \int_{-\infty}^{t} dt'\, F(X(t) - X(t'), t - t') + \Xi(X(t), t), \tag{16}$$

where the space- and time-dependent memory kernel $F$ is given by

$$F_j(x, t) = i\lambda^2 D \int \frac{d^d q}{(2\pi)^d} q_j q^2 |V_q|^2 e^{iq \cdot x - \alpha_q t}. \tag{17}$$

In this expression,

$$\alpha_q = Dq^2(q^2 + r), \tag{18}$$

is the relaxation rate of the field fluctuations with wavevector $q$, $V_q$ is the Fourier transform of $V(z)$, and the noise $\Xi(x, t)$ is Gaussian with vanishing average and correlation function

$$\langle \Xi_j(x, t)\Xi_l(x', t') \rangle = 2\gamma_0 T \delta_{jl} \delta(t - t') + T G_{jl}(x - x', t - t'), \tag{19}$$

where

$$G_{jl}(x, t) = \lambda^2 \int \frac{d^d q}{(2\pi)^d} q_j q_l \frac{|V_q|^2}{q^2 + r} e^{iq \cdot x - \alpha_q |t|}. \tag{20}$$

In passing we note that if the interaction potential $V(x)$ is invariant under spatial rotations (as assumed here), then $G_{jl} \propto \delta_{jl}$.

The equations (16)–(20) describing the effective dynamics of the probe are *exact*: the interaction with the Gaussian field introduces a space- and time-dependent memory term $F(x,t)$ and a position and time-dependent (non-Markovian) noise with correlation $G(x,t)$ in addition to the (Markovian) contribution $\propto \gamma_0$ in Eq. (19) due to the action of the thermal bath on the particle. In the next subsections we discuss first the relation between the non-linear memory $F(x,t)$ in Eq. (16) and the usual linear memory kernel $\Gamma(t)$ and then the fluctuation-dissipation relation which connects $F_j(x,t)$ to $G_{jl}(x,t)$.

## 3.2 The linear memory kernel

In order to relate the *non-linear* memory term $\propto F$ on the r.h.s. of Eq. (16) to the usual *linear* memory kernel $\Gamma(t)$ mentioned in the Introduction, we expand the former to the first order in the displacement $X(t) - X(t')$ and integrate by parts, obtaining

$$\int_{-\infty}^{t} dt' \, F_j(X(t) - X(t'), t - t') \simeq \int_{-\infty}^{t} dt' \left[ X_l(t) - X_l(t') \right] \nabla_l F_j(0, t - t') \tag{21}$$

$$= \int_{-\infty}^{t} dt' \dot{X}_l(t') \int_{0}^{\infty} dt'' \nabla_l F_j(0, t - t' + t'') \tag{22}$$

$$\equiv - \int_{-\infty}^{t} dt' \, \Gamma(t - t') \dot{X}_j(t'). \tag{23}$$

Here we have used the rotational invariance of $V(x)$ to get $F_j(0,t) = 0$ and $\nabla_l F_j(0,t) \propto \delta_{lj}$. In the last equation we identify the linear memory kernel $\Gamma(t)$ as given by

$$\int_{0}^{\infty} dt' \, \nabla_l F_j(0, t + t') = -\Gamma(t) \delta_{jl}. \tag{24}$$

Conversely, taking a linear function $F_j(x,t) = \dot{\Gamma}(t) x_j$ the expansion above is exact. Similarly, we also expand the noise correlation in Eq. (19) to the zeroth order in the displacement $x - x'$, and obtain

$$\langle \Xi_j(x,t) \Xi_l(x',t') \rangle \simeq 2\gamma_0 T \delta_{jl} \delta(t-t') + T G_{jl}(0, t-t'). \tag{25}$$

Using the explicit expression of $F(x,t)$ in Eq. (17) for calculating $\Gamma(t)$ from Eq. (24) and taking into account the definition of $G(x,t)$ in Eq. (20), one immediately gets

$$G_{jl}(0,t) = \Gamma(|t|) \delta_{jl}. \tag{26}$$

Accordingly, as expected, the total time-dependent friction $\gamma_0 \delta_+(t-t') + \Gamma(t-t')$, where $\delta_+(t)$ is the normalised delta-distribution on the half line $t > 0$, is related to the correlation of the noise in Eq. (25) by the standard fluctuation-dissipation relation involving the thermal energy $T$ as the constant of proportionality.

## 3.3 Fluctuation-dissipation relation

As expected, the fluctuation-dissipation relation discussed above for the linear approximation of the effective memory carries over to the non-linear case, still in a remarkably simple form which we derive here. In fact, the memory term $F_l(x,t)$ in Eq. (16) and the correlation $G_{jl}(x,t)$ of the noise in Eq. (19) are related, for $t > 0$, as

$$\nabla_j F_l(x,t) = \partial_t G_{jl}(x,t), \tag{27}$$

which is readily verified by using Eqs. (17) and (20). Beyond this specific case, in Appendix B we prove that the dynamics prescribed by Eqs. (16) and (19) is invariant under *time reversal* —and therefore the corresponding stationary distribution is an *equilibrium state*— if the non-linear dissipation and the correlation of the fluctuations satisfy Eq. (27).

The effective dynamics in Eq. (16), deriving from the coupling to a Gaussian field, is characterised by the memory kernel $F(x,t)$ which generalises the usual one $\Gamma(t)$ to the case of a non-linear dependence on $\dot{X}(t)$. In Sec. 4 we calculate the two-time correlation function in the presence of this non-linear memory kernel for a particle trapped in the harmonic potential with stiffness $\kappa$. Then we show that when the usual effective description of the dynamics of the particle in terms of a linear memory kernel is extracted from these correlation functions, this kernel turns out to depend on the stiffness $\kappa$, as observed in Refs. [8,9].

# 4 Perturbative calculation of the correlation functions

## 4.1 Perturbative expansion

In order to predict the dynamical behaviour of the probe particle, we need to solve the set of equations (6) and (7), which are made non-linear in $\{X, \phi\}$ by their coupling $\propto \lambda$. These non-linear equations are not solvable in general and thus we resort to a perturbative expansion in the coupling strength $\lambda$ by writing

$$\phi(x,t) = \sum_{n=0}^{\infty} \lambda^n \phi^{(n)}(x,t), \tag{28}$$

$$X(t) = \sum_{n=0}^{\infty} \lambda^n X^{(n)}(t), \tag{29}$$

where $\phi^{(0)}(x,t)$ and $X^{(0)}(t)$ are the solutions of Eqs. (6) and (7) for $\lambda = 0$, i.e., for the case in which the medium and the probe are completely decoupled. As usual, these expansions are inserted into Eqs. (6) and (7), which are required to be satisfied order by order in the expansion. In Refs. [13, 15, 16] this perturbative analysis was carried out within a path-integral formalism of the generating function of the dynamics of the system; below, instead, we present a direct calculation based on the perturbative analysis of the dynamical equations. We restrict ourselves to the quadratic order $n = 2$ of the expansion above and show that this is sufficient for capturing the non-trivial signatures of criticality as the correlation length $\xi$ of the fluctuations within the medium diverges. The validity of the perturbative approach will be discussed in Sec. 5, where our analytical predictions are compared with the results of numerical simulations.

As anticipated in Sec. 2.3, the motion of the probe particle in the absence of the coupling to the field is described by the Ornstein-Uhlenbeck process

$$\gamma_0 \dot{X}^{(0)}(t) = -\kappa X^{(0)}(t) + \zeta(t). \tag{30}$$

Its solution is

$$X^{(0)}(t) = \gamma_0^{-1} \int_{-\infty}^{t} ds\, \zeta(s) e^{-\omega_0(t-s)}, \tag{31}$$

where $\omega_0$ is the relaxation rate of the probe particle in the harmonic trap, introduced in Eq. (14). Hereafter we focus on the equilibrium behaviour of the colloid and therefore we assume that its (inconsequential) initial position $X(t = t_0) = X_0$ is specified at time $t_0 \to -\infty$.

The first two perturbative corrections $X^{(1)}(t)$ and $X^{(2)}(t)$ evolve according to the first-order differential equations

$$\dot{X}^{(n)} = -\omega_0 X^{(n)} + \gamma_0^{-1} f^{(n-1)}, \quad \text{with} \quad n = 1, 2, \tag{32}$$

where $f^{(0)} = f|_{\lambda=0}$ and $f^{(1)} = \mathrm{d}f/\mathrm{d}\lambda|_{\lambda=0}$ are obtained from the series expansion of $f$ in Eq. (8), in which the dependence on $\lambda$ is brought in implicitly by the expansions of $X$ and $\phi$.

Equations (32) are readily solved by

$$X^{(n)}(t) = \gamma_0^{-1} \int_{-\infty}^{t} \mathrm{d}s \, e^{-\omega_0(t-s)} f^{(n-1)}(s); \tag{33}$$

the forces $f^{(0)}$ and $f^{(1)}$ depend explicitly on the coefficients $\phi^{(n)}(x, t)$ of the expansion of the field $\phi(x, t)$ and thus, in order to calculate $X^{(1)}(t)$ and $X^{(2)}(t)$ above we need to know the time-evolution of the field. The latter is more conveniently worked out for the spatial Fourier transform

$$\phi_q(t) = \int \mathrm{d}^d x \, \phi(x, t) e^{iq \cdot x}, \tag{34}$$

of the field which, transforming Eq. (28), has the expansion

$$\phi_q(t) = \sum_{n=0}^{\infty} \lambda^n \phi_q^{(n)}(t). \tag{35}$$

In fact, from Eq. (6), one can write the time evolution

$$\dot{\phi}_q = -\alpha_q \phi_q + \lambda D q^2 V_q e^{iq \cdot X} + \eta_q, \tag{36}$$

where $\alpha_q$ is given in Eq. (18) and, as in Eqs. (17) and (20), $V_q$ is the Fourier transform of the interaction potential $V(x)$, while $\eta_q$ is the noise in Fourier space

$$\langle \eta_q(t) \eta_{q'}(t') \rangle = 2DT q^2 (2\pi)^d \delta^d(q + q') \delta(t - t'), \tag{37}$$

which is also delta-correlated in time.

For $\lambda = 0$, i.e., when the medium is decoupled from the probe, Eq. (36) is solved by

$$\phi_q^{(0)}(t) = \int_{-\infty}^{t} \mathrm{d}s \, \eta_q(s) e^{-\alpha_q(t-s)}, \tag{38}$$

where, as we are interested in the equilibrium behaviour, we assume hereafter that the (inconsequential) initial condition $\phi_q(t = t_0) = \phi_{q,0}$ is assigned at time $t_0 \to -\infty$. From this expression we can compute the two-time correlation function of the field $\phi_q^{(0)}(t)$:

$$\left\langle \phi_q^{(0)}(t) \phi_{q'}^{(0)}(t') \right\rangle = (2\pi)^d \delta^d(q + q') \mathcal{G}_q(t - t'), \tag{39}$$

where we introduced

$$\mathcal{G}_q(t) = \frac{DT q^2}{\alpha_q} e^{-\alpha_q |t|}. \tag{40}$$

The dynamics of the linear correction $\phi_q^{(1)}$ can be obtained from Eq. (36), after inserting Eqs. (29) and (35), by comparing the coefficients of order $\lambda$ on both sides, finding

$$\dot{\phi}_q^{(1)} = -\alpha_q \phi_q^{(1)} + D q^2 V_q \, e^{iq \cdot X^{(0)}(t)}, \tag{41}$$

which is solved by

$$\phi_q^{(1)}(t) = Dq^2 V_q \int_{-\infty}^{t} ds \ e^{-\alpha_q(t-s)} e^{iq \cdot X^{(0)}(s)}. \tag{42}$$

Accordingly, $\phi_q^{(1)}(t)$ depends on $X^{(0)}(t)$ which, in turn, affects $X^{(2)}(t)$. As we will see below, for calculating the correlation function $C(t)$ of the particle coordinate up to order $\lambda^2$ it is sufficient to know the evolution of $\phi_q^{(0)}(t)$ and $\phi_q^{(1)}(t)$.

## 4.2 Two-time correlation function

We now compute perturbatively the two-time correlation $C(t)$ by expanding it in a series in $\lambda$. First, we note that $C$ is expected to be an even function of $\lambda$ because of the invariance of the equations of motion under the transformation (9) and thus the correction of order $\lambda$ vanishes, leading to

$$C(t) = C^{(0)}(t) + \lambda^2 C^{(2)}(t) + \mathcal{O}(\lambda^4), \tag{43}$$

where $C^{(0)}$ is the auto-correlation of the free colloid reported in Eq. (13).

As detailed in Appendix C, the correction $C^{(2)}$ is given by

$$C^{(2)}(t) = d \left[ \left\langle X_j^{(1)}(t) X_j^{(1)}(0) \right\rangle + \left\langle X_j^{(0)}(t) X_j^{(2)}(0) \right\rangle + \left\langle X_j^{(2)}(t) X_j^{(0)}(0) \right\rangle \right], \tag{44}$$

for a generic value of $j \in \{1, 2, \cdots, d\}$ (we assume rotational symmetry of the problem). Using Eqs. (31) and (33), one can readily calculate the three contributions above, as reported in detail in Appendix C. The final result can be expressed as

$$C^{(2)}(t) = \frac{DT}{\gamma_0^2} \int_0^t du \ e^{-\omega_0(t-u)} (t-u) \mathcal{F}(u) \quad \text{for} \quad t > 0, \tag{45}$$

where

$$\mathcal{F}(u) = \int \frac{d^d q}{(2\pi)^d} \frac{q^4}{\alpha_q} |V_q|^2 \exp\left(-\alpha_q u - \frac{q^2 T}{\kappa}\left[1 - e^{-\omega_0 u}\right]\right). \tag{46}$$

The correlation function $C^{(2)}(t)$ for $t < 0$ can be obtained from Eq. (45) by using the fact that, in equilibrium, $C^{(2)}(-t) = C^{(2)}(t)$. Equations (45) and (46) constitute the main predictions of this analysis. We anticipate here that in Sec. 4.5 we prove that the function $\mathcal{F}(t)$ introduced above is, up to an inconsequential proportionality constant, the correction to the effective linear memory kernel due to the coupling of the particle to the bath and therefore it encodes the emergence of a non-Markovian dynamics, as discussed below [see, in particular, Eq. (49)].

The expressions in Eqs. (45) and (46), after a rescaling of time in order to measure it in units of $\omega_0^{-1}$, is characterised by the emergence of a dimensionless combination $\alpha_q/\omega_0 = Dq^2(q^2 + r)/\omega_0$ in the exponential. In turn, the dependence of this factor on $q$ defines, as expected, a length scale $r^{-1/2}$ — actually corresponding to the correlation length $\xi$ of the fluctuations of the field — and $\ell_D = (D/\omega_0)^{1/4} = (D\gamma_0/\kappa)^{1/4}$. This latter scale corresponds to the typical spatial extent of the fluctuations of the field which relax on the typical timescale $\omega_0^{-1}$ of the relaxation of the particle in the trap.

The resulting dynamical behavior of the system is therefore characterised by the length-scales $\ell_T$ (see after Eq. (14)), $\ell_D$ introduced above, $\xi$, the colloid radius $R$ and by the timescales associated to them. We shall see below, however, that the complex interplay between these scales simplifies in some limiting cases in which they become well-separated and universal expressions emerge.

## 4.3 Long-time algebraic behaviours of correlations

The dynamics of the particle coupled to the fluctuating field naturally depends also on the details of their mutual interaction potential $V(x)$. However, as we shall show below, some aspects of the dynamics acquire a certain degree of universality, as they become largely independent of the specific form of $V(x)$. In particular, we focus on the experimentally accessible two-time correlation function $C(t)$ of the probe position and consider its long-time decay. From Eq. (45) it can be shown (see Appendix C.3 for details) that, at long times $t \gg \omega_0^{-1}$, the leading-order correction $C^{(2)}$ to the correlation function [see Eq. (43)] in the stationary state, is given by

$$C^{(2)}(t \gg \omega_0^{-1}) \simeq \frac{DT}{\kappa^2} \mathcal{F}(t), \tag{47}$$

while $C^{(0)}(t)$, the contribution from the decoupled dynamics, is generically an exponentially decaying function of $t$, given by Eq. (13) in the stationary state. For large $t$, the integral over $q$ in $\mathcal{F}$ [see Eq. (46)] turns out to be dominated by the behaviour of the integrand for $q \to 0$ and, at the leading order, it can be written as

$$\mathcal{F}(t) \simeq V_0^2 \int \frac{d^d q}{(2\pi)^d} \frac{q^4}{\alpha_q} e^{-\alpha_q t}, \tag{48}$$

where we used the fact that $|V_q|^2 = V_0^2 + O(q^2)$ and $\exp[-q^2(T/\kappa)(1-e^{-\omega_0 t})] = 1 + O(q^2)$ for $q \to 0$. It is then easy to see that Eq. (48) takes the scaling form

$$\mathcal{F}(t) = (V_0^2/D)(Dt)^{-d/4} \mathcal{G}(r\sqrt{Dt}), \tag{49}$$

with the dimensionless scaling function

$$\mathcal{G}(w) = \frac{\Omega_d}{2(2\pi)^d} \int_0^\infty dz \frac{z^{d/2}}{z+w} e^{-z(z+w)}, \tag{50}$$

where $\Omega_d = 2\pi^{d/2}/\Gamma(d/2)$ denotes the solid angle in $d$-dimensions. Clearly, for $w \to 0$, the scaling function approaches a constant value $\mathcal{G}(0) = \Omega_d \Gamma(d/4)/[4(2\pi)^d] = \Gamma(d/4)/[2(4\pi)^{d/2}\Gamma(d/2)]$. In the opposite limit $w \gg 1$, instead, the factor $e^{-zw}$ in the integrand of Eq. (50) allows us to expand the remaining part of the integrand for $z \to 0$ and eventually find $\mathcal{G}(w \to \infty) = d[\Gamma(d/2)/\Gamma(d/4)]\mathcal{G}(0)w^{-(2+d/2)}[1+\mathcal{O}(w^{-2})]$. Accordingly, the long-time decay of the correlation function shows two dynamical regimes:

$$C^{(2)}(t) \propto \begin{cases} t^{-d/4} & \text{for} \quad Dt \ll r^{-2}, \\ t^{-(1+d/2)} & \text{for} \quad Dt \gg r^{-2}. \end{cases} \tag{51}$$

At the critical point $r = 0$ of the fluctuating medium, the second regime above cannot be accessed and only an algebraic decay $\propto t^{-d/4}$ is observed. For any finite value $r > 0$, instead, there is a crossover from the critical-like behaviour $\propto t^{-d/4}$ to the off-critical and faster decay $\propto t^{-(1+d/2)}$ as the time $t$ increases beyond the time-scale $\propto r^{-2}/D$. The emergence of an algebraic decay of correlations away from criticality is due to the presence of the local conservation law in the dynamics of the field. The time scale at which this crossover occurs is expected to be given by $\sim \xi^z$ where $\xi$ is the correlation length and $z = 4$ is the dynamical critical exponent of the Gaussian medium with the conserved dynamics we are considering here. Remembering that $\xi \sim r^{-\nu}$, with $\nu = 1/2$ in this case, a scaling collapse of the different curves corresponding to $r > 0$ is expected when plotted as a function of $t/\xi^z \sim tr^2$, as in Eq. (49). Given that the $\lambda$-independent contribution $C^{(0)}$ to the stationary autocorrelation function is characterised by an exponential decay with typical time scale $\omega_0^{-1}$, Eq. (51) shows that the

long-time behaviour of the total correlation function $C(t)$ is actually completely determined by the slow algebraic decay of $\lambda^2 C^{(2)}(t)$ as soon as $t \gg \omega_0^{-1}$ and thus the effect of the coupling to the field can be easily revealed from the behaviour of the correlation function at long times. As we also discuss further below in Sec. 4.6, Eq. (47) implies that $C^{(2)}(t)$ inherits the algebraic long-time behaviour from $\mathcal{F}(t)$, which is solely determined by the dynamics of the field. In fact, from Eqs. (49) and (47) we see that the actual dynamical properties of the particle (i.e., its drag coefficient $\gamma_0$), the form of the interaction potential $V$, the temperature $T$ of the bath and the parameter $\kappa$ of the trapping potential determine only the overall amplitude of $C^{(2)}(t \gg \omega_0^{-1})$.

Figure 2 shows plots of $C^{(2)}(t)$ evaluated using Eqs. (45) and (46) with a Gaussian-like interaction potential

$$V(z) = \varepsilon\, e^{-z^2/(2R^2)}, \tag{52}$$

(which can be seen as modelling a colloid of "radius" $R$ and typical interaction energy $\varepsilon = 1$) in spatial dimension $d = 1, 2$, and $3$, from top to bottom, and for various values of $r$. Depending on the value of $r$ we observe the crossover described by Eq. (51), which is highlighted in the insets showing the data collapse according to Eq. (49).

In summary, the equilibrium correlation $\langle X(0) \cdot X(t) \rangle$ of the position $X$ of a particle with overdamped dynamics, diffusing in a harmonic trap and in contact with a Gaussian field with conserved dynamics, shows an algebraic decay at long times, which is *universal* as it is largely independent of the actual form of the interaction potential and of the trapping strength but depends only on the spatial dimensionality $d$. At the leading order in coupling strength $\lambda$, one finds

$$C(t) = \langle X(0) \cdot X(t) \rangle \propto \lambda^2 \begin{cases} t^{-d/4} & \text{at } r = 0, \\ t^{-(1+d/2)} & \text{for } r > 0. \end{cases} \tag{53}$$

The exponents of these algebraic laws, as well as those discussed further below, reflect the mean-field critical exponents which characterise the static and dynamical behavior of the Gaussian field considered here.

## 4.4 Power spectral density

In the experimental investigation of the dynamics of tracers in various media, one is naturally lead to consider the power spectral density (PSD) $S(\omega)$ of the trajectory $X(t)$, which is defined as the Fourier transform of the correlation $C(t)$:

$$S(\omega) = \int_{-\infty}^{+\infty} dt\, C(t) e^{i\omega t} = S^{(0)}(\omega) + \lambda^2 S^{(2)}(\omega) + \mathcal{O}(\lambda^4). \tag{54}$$

The zeroth-order term, corresponding to the Ornstein-Uhlenbeck process, was discussed above and is given by Eq. (15). The second-order correction $S^{(2)}(\omega)$ can be obtained by Fourier transforming Eq. (45) (see Appendix C for details), which yields

$$S^{(2)}(\omega) = \frac{2DT}{\gamma_0^2(\omega_0^2 + \omega^2)^2} \int_0^{+\infty} du\left[\left(\omega_0^2 - \omega^2\right)\cos(\omega u) - 2\,\omega\omega_0\,\sin(\omega u)\right]\mathcal{F}(u), \tag{55}$$

in terms of $\mathcal{F}(u)$ given in Eq. (46).

Figure 3 shows the behaviour of the PSD for the Gaussian interaction potential in Eq. (52). In panel (a) the resulting $S(\omega)$ at order $\lambda^2$ is plotted as a function of $\omega$ for various values of $r$. The large-$\omega$ behavior of $S(\omega)$ is essentially determined by $S^{(0)}(\omega)$ in Eq. (15) and therefore it is dominated by the Brownian thermal noise $\zeta(t)$ in Eq. (3) with $S(\omega) \simeq 2dT/(\gamma_0\omega^2)$ [see also Eq. (14)]. Heuristically, at sufficiently high frequencies the dynamics of the particle

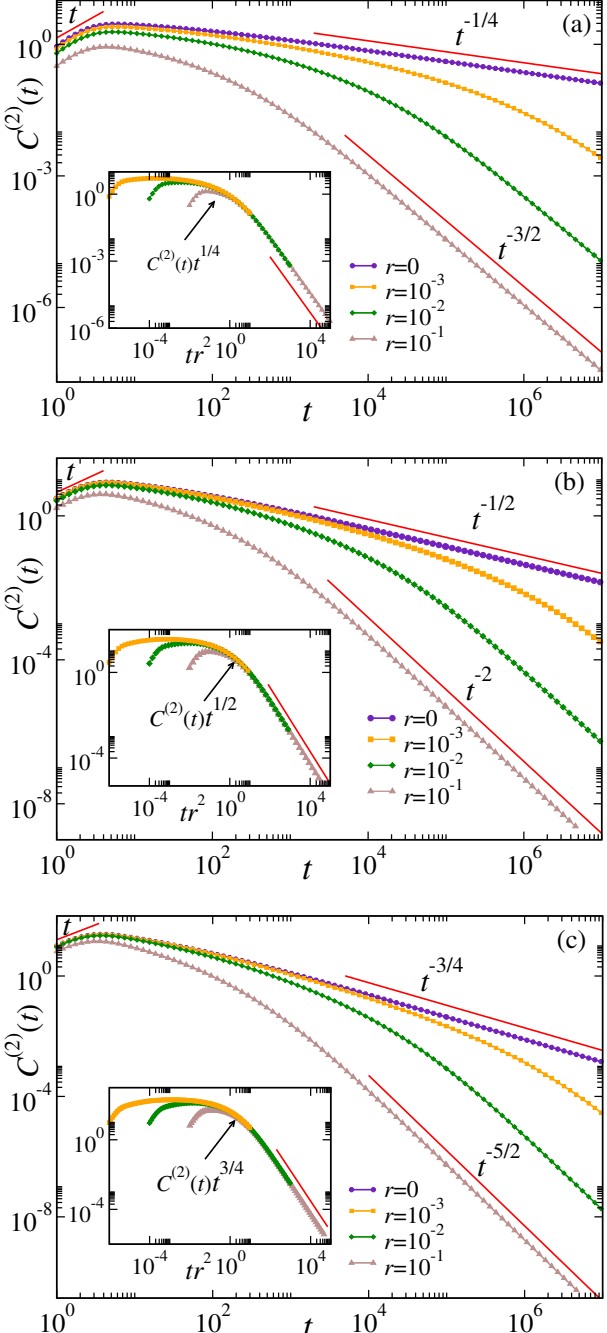

Figure 2: Stationary auto-correlation function $C^{(2)}(t)$ of the position of the colloidal particle, as obtained numerically in spatial dimension (a) $d = 1$, (b) 2, and (c) 3, for various values of the parameter $r$ which controls the spatial and temporal extents of the correlations of the fluctuations within the medium. At the critical point $r = 0$, the correlation $C^{(2)}(t)$ decays as $t^{-d/4}$ at long times, while away from the critical point $C^{(2)}(t) \sim t^{-(1+d/2)}$. These slow algebraic decays — indicated in the various panels by the straight lines on the right — determine that of the total auto-correlation function $C(t)$, as the contribution of order $\lambda^0$ decays exponentially upon increasing time. At short times, instead, $C^{(2)}(t)$ turns out to grow linearly upon increasing $t$, as indicated by the solid lines on the left of each panel and as discussed, c.f., in Sec. 4.6. The insets show the scaling collapse of $t^{d/4}C_2(t)$ when the same data is plotted as a function of $t\,r^2$. Here we considered an interaction potential $V(z) = e^{-z^2/(2R^2)}$ with $R = 2$ and $D = T = 1$ and $\kappa = \gamma_0 = 1$.

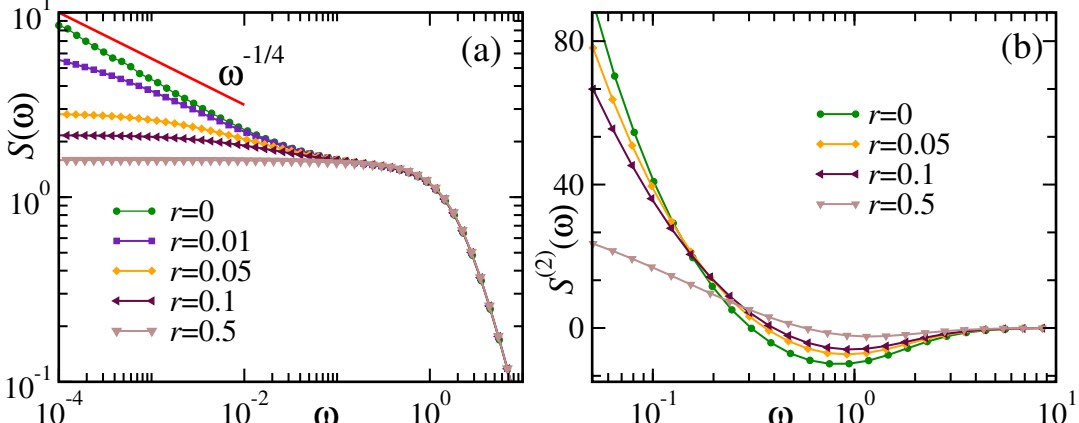

Figure 3: (a) Structure factor $S(\omega) = S^{(0)}(\omega) + \lambda^2 S^{(2)}(\omega)$ [see Eqs. (15) and (55)] and (b) leading-order correction $S^{(2)}(\omega)$ as functions of $\omega$ for a set of values of $r$ and $\lambda = 0.05$ in spatial dimension $d = 3$. At criticality, (i.e., for $r = 0$) the correction $S^{(2)}(\omega)$ (and therefore $S(\omega)$) displays the algebraic singularity predicted by Eq. (57) for $\omega \to 0$, which is indicated by the solid line in panel (a). Remarkably, $S^{(2)}(\omega)$ becomes negative upon increasing $\omega$ above a certain $r$-dependent value. Here we have used a Gaussian interaction potential $V(z) = e^{-z^2/(2R^2)}$ with $R = 2$. The other parameters are $\gamma_0 = T = D = 1$ with $\kappa = 2$.

effectively decouples from the relatively slow dynamics of the field $\phi$ and therefore its behavior is independent of $\lambda$. Correspondingly, one indeed observes that $S^{(2)}(\omega)$ reported in panel (b) vanishes as $\omega$ increases, independently of the specific value of the parameter $r$, i.e., of the correlation length $\xi$ of the fluctuating medium. In particular, in Appendix D.2 it is shown that $S^{(2)}(\omega) \propto \omega^{-4}$ for large $\omega$ and that $S^{(2)}(\omega)$ approaches zero from below. On the contrary, the coupling to the field becomes relevant at lower frequencies and it increases as $r \to 0$, i.e., upon approaching the critical point of the fluctuating field, as shown in panel (b) of Fig. 3, which displays the pronounced contribution of $S^{(2)}(\omega)$ as $\omega \to 0$. This contribution is discussed in detail in Appendix D by studying the asymptotic behavior of Eq. (55) but it can be easily understood from the stationary autocorrelation at long times in Eq. (51), from which we can infer the behaviour of $S^{(2)}(\omega)$ as $\omega \to 0$, which approaches the generically finite value [where we used Eq. (14)]

$$S^{(2)}(\omega = 0) = \frac{2DT}{\kappa^2} \int_0^{+\infty} du\, \mathcal{F}(u). \tag{56}$$

In particular, by using Eq. (51), we find that $S^{(2)}(\omega)$ is actually finite as $\omega \to 0$ for $r > 0$, i.e., away from criticality or generically for $d \geq 4$, while

$$S^{(2)}(\omega \to 0) \propto \omega^{-1+d/4} \quad \text{for} \quad r = 0 \quad \text{and} \quad d < 4. \tag{57}$$

(See Eq. (139) in Appendix D.2 for the complete expression, including the coefficients of proportionality.)

Correspondingly, given that the zeroth-order contribution $S^{(0)}(\omega = 0)$ to $S(\omega)$ is finite as $\omega \to 0$, the power spectral density for a particle in a critical medium develops an integrable algebraic singularity as $\omega \to 0$ which is entirely due to the coupling to the medium.

Note that, according to the discussion in Sec. 2.2, the equal-time correlation $C(0)$ of the position of the probe particle is not influenced by its coupling to the field, i.e., it is independent of $\lambda$. In turn, these equal-time fluctuations are given by the integral of $S(\omega)$ over $\omega$ and therefore one concludes that the Fourier transform $S^{(n)}(\omega)$ of the contributions $C^{(n)}(t)$ of

order $\lambda^n$ in the generalisation of the expansion in Eq. (43), have to satisfy

$$\int_0^{+\infty} d\omega\, S^{(n)}(\omega) = 0 \quad \text{for} \quad n \neq 0. \tag{58}$$

Accordingly, the integral on the linear scale of the curves in panel (b) of Fig. 3 has to vanish, implying that the positive contribution at small $\omega$ is compensated by the negative contribution at larger values, which is clearly visible in the plot and which causes a non-monotonic dependence of $S^{(2)}(\omega)$ on $\omega$.

## 4.5 Effective memory kernel

As we mentioned in the introduction, the most common modelling of the overdamped dynamics of a colloidal particle in a medium is done in terms of a *linear* evolution equation with an effective memory kernel of the form (see also the discussion in Sec. 3.3)

$$\int_{-\infty}^{t} dt'\, \Gamma(t-t')\dot{X}_j(t') = \tilde{f}_j(t) + \tilde{\zeta}_j(t), \tag{59}$$

where

$$\langle \tilde{\zeta}_j(t)\tilde{\zeta}_l(t')\rangle = 2T\, \delta_{jl}\Gamma(|t-t'|), \tag{60}$$

and $\tilde{f}$ represents the forces acting on the colloid in addition to the stochastic noise $\tilde{\zeta}$ provided by the effective equilibrium bath at temperature $T$, for which the fluctuation-dissipation relation is assumed to hold. An additional, implicit assumption of the modelling introduced above is that $\Gamma$ describes the effect of the fluctuations introduced by the thermal bath, which are expected to be independent of the external forces $\tilde{f}$. We shall see below that this is not generally the case because of the intrinsic non-linear nature of the actual evolution equation. In the setting we are interested in, the external force $\tilde{f}$ in Eq. (59) is the one provided by the harmonic trap, i.e., $\tilde{f}(t) = -\kappa X(t)$. In this case, the process is Gaussian and the Laplace transform $\hat{C}(p)$ of the stationary correlation function $C(t)$ is given by [9, 27]

$$\hat{C}(p) = \frac{dT\hat{\Gamma}(p)}{\kappa[\kappa + p\hat{\Gamma}(p)]}, \tag{61}$$

in terms of the Laplace transform $\hat{\Gamma}(p)$ of $\Gamma(t)$. As it is usually done in microrheology [4], we invert this relation in order to infer the memory kernel from the correlations, i.e.,

$$\hat{\Gamma}(p) = \frac{\kappa\hat{C}(p)}{dT/\kappa - p\hat{C}(p)}. \tag{62}$$

The perturbative expansion of $C(t)$ in Eq. (43) readily translates into a similar expansion for the corresponding Laplace transform, i.e.,

$$\hat{C}(p) = \hat{C}^{(0)}(p) + \lambda^2\hat{C}^{(2)}(p) + \mathcal{O}(\lambda^4), \tag{63}$$

where the Laplace transform of the correlation for $\lambda = 0$ in Eq. (13) and of the correction in Eq. (45) can be easily calculated:

$$\hat{C}^{(0)}(p) = \frac{dT}{\kappa(p+\omega_0)}, \tag{64}$$

$$\hat{C}^{(2)}(p) = \frac{DT}{\gamma_0^2}\frac{\hat{\mathcal{F}}(p)}{(p+\omega_0)^2}. \tag{65}$$

Inserting the perturbative expansion (63) in Eq. (62) with $\hat{C}^{(0)}(p)$ and $\hat{C}^{(2)}(p)$ given above, and using Eq. (14), we obtain the corresponding perturbative expansion for the memory kernel

$$\hat{\Gamma}(p) = \hat{\Gamma}^{(0)}(p) + \lambda^2 \hat{\Gamma}^{(2)}(p) + \mathcal{O}(\lambda^4) = \gamma_0 + \frac{\lambda^2 D}{d} \hat{\mathcal{F}}(p) + \mathcal{O}(\lambda^4). \tag{66}$$

From this expression, after transforming back in the time domain, we naturally find

$$\Gamma(t) = \Gamma^{(0)}(t) + \lambda^2 \Gamma^{(2)}(t) + \mathcal{O}(\lambda^2), \tag{67}$$

in which we recover the expected memory kernel

$$\Gamma^{(0)}(t) = \gamma_0 \delta_+(t), \tag{68}$$

in the absence of the interaction with the field — which renders Eq. (30) — and the correction

$$\Gamma^{(2)}(t) = \frac{D}{d} \mathcal{F}(t), \tag{69}$$

due to this interaction. This equality provides also a simple physical interpretation of the function $\mathcal{F}(t)$ introduced in Eqs. (45) and (46) as being the contribution to the linear friction due to the interaction of the particle with the field.

This effective memory $\Gamma^{(2)}(t) \propto \mathcal{F}(t)$ turns out to depend explicitly on the stiffness $\kappa$ of the trap, as prescribed by Eq. (46). Contrary to the very same spirit of writing an equation such as Eq. (59), the effective memory $\Gamma(t)$ is not solely a property of the fluctuations of the medium, but it actually turns out to depend on all the parameters which affect the dynamics of the probe, including the external force $\tilde{f}$. This dependence is illustrated in Fig. 4 which shows $\Gamma^{(2)}(t)$ as a function of time $t$ for various values of the relevant parameters. In particular, the curves in panel (a) correspond to a fixed value of the parameter $r$ and shows the dependence of the correction $\Gamma^{(2)}(t)$ on the trap stiffness $\kappa$, while panel (b) shows the dependence on the distance $r$ from criticality (equivalently, on the spatial range $\xi = r^{-1/2}$ of the correlation of the field) for a fixed value of $\kappa$. The curves in panel (a) show that, depending on the value of $\kappa$, a crossover occurs between the behaviour at short times — during which the particle does not displace enough to experience the effects of being confined — corresponding to a weak trap with $\kappa \to 0$ and that at long times $t \gg \omega_0^{-1}$ corresponding to the strong-trap limit $\kappa \to \infty$, which is further discussed below in Sec. 4.6. In turn, as shown by panel (b), the power of the algebraic decay of $\Gamma^{(2)}(t)$ at long times $t \gg \omega_0^{-1}$ depends on whether the field is critical ($r = 0$) or not ($r \neq 0$). In particular, upon increasing $t$ one observes, after a faster relaxation at short times controlled, inter alia, by the trap stiffness $\kappa$, a crossover between a critical-like slower algebraic decay and a non-critical faster decay, with the crossover time diverging as $\sim r^{-2}$ for $r \to 0$. Taking into account that, at long times $t \gg \omega_0^{-1}$, Eqs. (47) and (69) imply that $\Gamma^{(2)}(t)$ is proportional to $C^{(2)}(t)$ according to

$$\Gamma^{(2)}(t \gg \omega_0^{-1}) = \frac{\kappa^2}{dT} C^{(2)}(t), \tag{70}$$

this crossover is actually the one illustrated in Fig. 2 for $C^{(2)}(t)$ in various spatial dimensions $d$.

We emphasise that the long-time behaviour of the correlation $\Gamma(t \gg \omega_0^{-1})$ of the effective fluctuating force generated by the near-critical medium (according to Eqs. (59) and (60)) and acting on the probe particle is actually independent of the trapping strength $\kappa$ and is characterised by an algebraic decay as a function of time, following from Eqs. (70) and (51). In particular, in spatial dimension $d = 3$, this decay is $\propto t^{-3/4}$ at criticality and $\propto t^{-5/2}$ for the non-critical case, with a *positive* coefficient of proportionality. This correlated effective

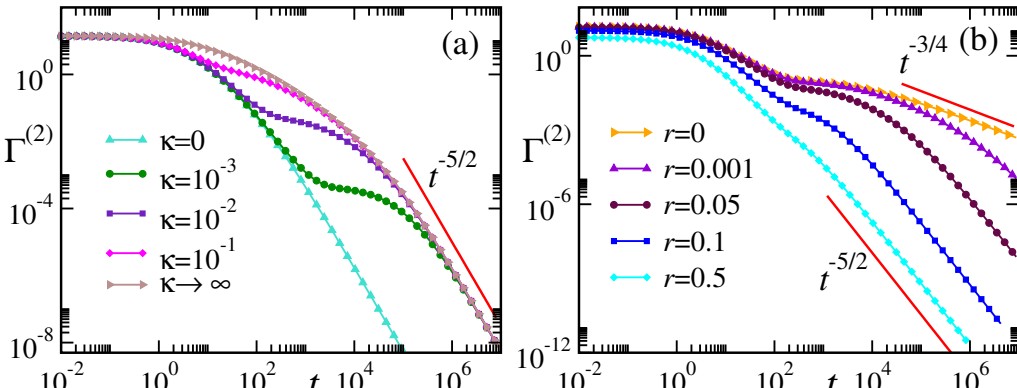

Figure 4: Effective memory kernel $\Gamma^{(2)}(t)$ in spatial dimension $d = 3$ obtained from Eqs. (69) and (46) via a numerical integration. Panel (a) shows $\Gamma^{(2)}(t)$ for various values of the trap stiffness $\kappa$ and fixed $r = 0.01$ while panel (b) for various values of $r$ and fixed $\kappa = 0.01$. The curves corresponding to the limiting cases $\kappa \to \infty$ and $\kappa = 0$ reported in panel (a) are obtained from Eq. (69) by using the approximate expressions of $\mathcal{F}$ in, c.f., Eqs. (75) and (78), respectively. The algebraic asymptotic decays indicated in panels (a) and (b), instead, follow from Eq. (70) and Fig. 2 (see also Eq. (51)). Here we have used the interaction potential $V(z) = e^{-z^2/(2R^2)}$ with $R = 2$, while the other parameters of the model are $\gamma_0 = D = T = 1$.

force can be compared with the one emerging on a Brownian particle due to hydrodynamic memory, generated by the fluid medium backflow, which turns out to have also algebraic correlations, with decay $\propto t^{-3/2}$ and a *negative* coefficient of proportionality, characteristic of anticorrelations (see, e.g., Ref. [2]). Accordingly, the algebraic decay of the hydrodynamic memory is faster than that due to the field in the critical case but slower than the one observed far from criticality. As an important additional qualitative difference between these two kinds of effective correlated forces, while the long-time anticorrelations due to the hydrodynamic memory may give rise to resonances in $S(\omega)$ [2], this is not the case for the effective force due to the coupling to the field.

The dependence of the effective memory $\Gamma^{(2)}$ on the trapping strength $\kappa$ discussed above carries over to the friction coefficient

$$\gamma = \int_0^\infty \mathrm{d}t\, \Gamma(t) = \hat{\Gamma}(p = 0) = \gamma_0 + \lambda^2 \gamma^{(2)} + \mathcal{O}(\lambda^4), \tag{71}$$

which is usually measured in experimental and numerical studies [8]. In the last equality we used Eqs. (67) and (68). Using, instead, Eq. (62) for $p \to 0$ and the relationship $\hat{C}(p = 0) = S(\omega = 0)/2$ between the Laplace and the Fourier transform of the two-time correlation function, one finds the following relationship between $\gamma$ and $S(\omega)$:

$$\gamma = \frac{\kappa^2}{2dT} S(\omega = 0). \tag{72}$$

Accordingly, the correction $\gamma^{(2)}$ to $\gamma$ in Eq. (71), due to the coupling to the field is, up to a constant, equivalently given by the integral of $\Gamma^{(2)}(t)$ in Eq. (69) or by $S^{(2)}(\omega = 0)$ in Eq. (56). In Fig. 5 we show the dependence of $\gamma^{(2)}$ on the trap stiffness $\kappa$, for a representative choice of the various parameters and upon approaching the critical point (i.e., upon decreasing $r$) from bottom to top. In particular, by using Eq. (72) and the results of Appendix D.2 (see, c.f., Eqs. (136) and (137)), one finds that, in the limit of weak trapping,

$$\gamma^{(2)}(\kappa \to 0; r \to 0) \simeq \frac{\gamma_0 |V_0|^2}{T} r^{-1+d/2} \frac{\Gamma(1-d/2)}{d(4\pi)^{d/2}} \quad \text{for} \quad d < 2, \tag{73}$$

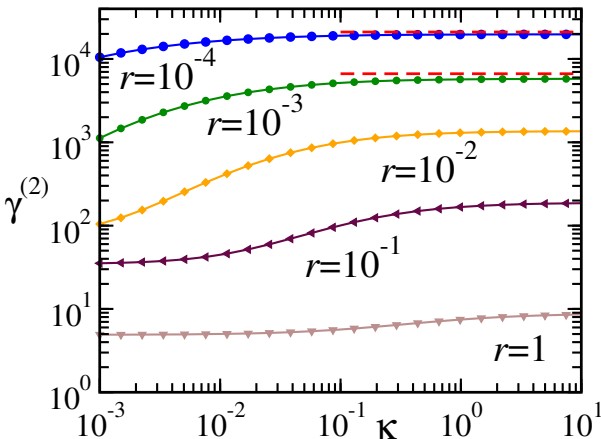

Figure 5: Dependence of the correction $\gamma^{(2)}$ to the friction coefficient $\gamma$ (see Eqs. (71) and (72)) on the trap stiffness $\kappa$ for various values of $r$, decreasing from top to bottom, in spatial dimension $d = 3$. The horizontal dashed lines for $r = 10^{-3}$ and $10^{-4}$ correspond to the asymptotic behavior given by Eq. (74). The interaction potential used for this plot is $V(z) = e^{-z^2/(2R^2)}$ with $R = 2$ and the other parameters are $\nu = D = T = 1$.

grows with an algebraic and *universal* singularity as a function of $r$ for $r \to 0$ while, for strong trapping,

$$\gamma^{(2)}(\kappa \to \infty; r \to 0) \simeq \frac{|V_0|^2}{D} r^{-2+d/2} \frac{\Gamma(2-d/2)}{d(4\pi)^{d/2}} \quad \text{for} \quad d < 4, \tag{74}$$

grows with a different exponent, with the previous limits being otherwise finite. Accordingly, for $d < 4$ one observes generically that the values of $\gamma^{(2)}$ for $\kappa \to 0$ and $\kappa \to \infty$ grow upon approaching criticality, with the latter growing more than the former, as clearly shown in Fig. 5. The corresponding increase of the friction coefficient upon increasing the stiffness was already noted in Ref. [16] for the same model, and resembles the one found in molecular dynamics simulations of a methane molecule in water (compare with Fig. 4 of Ref. [8]).

## 4.6 Limits of strong and weak confinement

Here, we specialise the analysis presented above to the strong- and weak-trap limits, formally corresponding to $\kappa \to \infty$ and $\kappa \to 0$, respectively. Note that, while the system cannot reach an equilibrium state for $\kappa = 0$ due to the diffusion of the particle, the limit $\kappa \to 0$ of the various quantities calculated in such a state for $\kappa \neq 0$ is well-defined and, as suggested by Fig. 4, it describes the behaviour of the system at short and intermediate time scales. The strong-trap limit, instead, captures the behaviour of the particle at times $t \gg \omega_0^{-1}$ and it corresponds to a probe that is practically pinned at the origin, with a small displacement that is proportional to the force exerted on it by the field. In this limit, the timescale $\omega_0^{-1}$ of the relaxation in the trap [see Eq. (14)] is small compared to all the other timescales. The function $\mathcal{F}$ in Eq. (46) then reduces to

$$\mathcal{F}(t) \simeq \int \frac{d^d q}{(2\pi)^d} \frac{q^4}{\alpha_q} |V_q|^2 \exp\left(-\alpha_q t\right). \tag{75}$$

At long times, this expression behaves as the one Eq. (48), i.e., as in Eqs. (49) and (50). Equation (75), via Eq. (69), determines also the limiting behaviour of $\Gamma^{(2)}(t)$ reported in panel (a) of Fig. 4, which exhibits at long times the crossover predicted by Eq. (49), shown in panel

(b) of the same figure. Similarly, the correction $C^{(2)}(t)$ to the correlation $C(t)$ is then given by Eq. (47), i.e.,

$$C^{(2)}(t) \simeq \frac{DT}{\kappa^2} \int \frac{d^d q}{(2\pi)^d} \frac{q^4}{\alpha_q} |V_q|^2 \exp\left(-\alpha_q t\right), \tag{76}$$

which, as expected, vanishes as $\propto \kappa^{-2}$ in the limit $\kappa \to \infty$, corresponding to $X \propto \kappa^{-1}$. However, this correction reflects the fluctuations of the force $f(X, \phi(x))$ exerted by the field on the probe, given by Eq. (8): the particle being practically pinned at $X = 0$, the correlations of the force $f \simeq \kappa X/\lambda$ are

$$\langle f(0, \phi(x, t)) \cdot f(0, \phi(x, 0)) \rangle = \kappa^2 C^{(2)}(t). \tag{77}$$

Following Sec. 4.3, one can show that the correlations (76) still exhibit an algebraic decay at long times, as the one of Eq. (48). This means that the long-time algebraic behaviour of the correlations of the position of the particle is actually a property of the field itself and it does not come from the interplay between the particle and the field dynamics, although it might depend on the form of the coupling between the particle and the medium.

In the weak-trap limit $\kappa \to 0$ (i.e., $\omega_0 \to 0$) the function $\mathcal{F}$ in Eq. (46) becomes

$$\mathcal{F}(t) \simeq \int \frac{d^d q}{(2\pi)^d} \frac{q^4}{\alpha_q} |V_q|^2 \exp\left(-(\alpha_q + q^2 T/\gamma_0)t\right), \tag{78}$$

which determines, via Eq. (69), the behavior of $\Gamma^{(2)}(t)$ in the same limit, shown in panel (a) of Fig. 4. The exponential in this expression shows the natural emergence of the length scale $\ell \equiv (D\gamma_0/T)^{1/2}$ influencing the dynamics even at long times, when other length scales such as $R$ turn out to be irrelevant. This scale can actually be expressed as the only $\kappa$-independent combination of the $\kappa$-dependent scales $\ell_{T,D}$ discussed after Eq. (46), i.e., $\ell = \ell_D^2/\ell_T$. In particular, at long times and sufficiently close to criticality such that $r \ll \ell^{-2}$, Eq. (78) takes the scaling form

$$\mathcal{F}(t) \simeq (V_0^2/D)\ell^d (Dt)^{-d/2} \mathcal{G}_{\text{wt}}(rDt/\ell^2), \tag{79}$$

with the dimensionless scaling function

$$\mathcal{G}_{\text{wt}}(w) = \frac{\Omega_d}{2(2\pi)^d} \int_0^\infty dz \frac{z^{d/2}}{z+w} e^{-z}, \tag{80}$$

where $\Omega_d$ is the solid angle reported after Eq. (50). For $w \to 0$ the scaling function renders $\mathcal{G}_{\text{wt}}(0) = \Omega_d \Gamma(d/2)/[2(2\pi)^d] = (4\pi)^{-d/2}$. In the opposite limit $w \gg 1$, instead, one has $\mathcal{G}_{\text{wt}}(w \to \infty) = \Omega_d \Gamma(d/2+1)/[2(2\pi)^d]w^{-1} = d/[2(4\pi)^{d/2}]w^{-1}$. At sufficiently long times (but still within the range of validity of the weak-trap approximation), one thus finds that $\mathcal{F}(t) \propto t^{-(1+d/2)}$ for $r \neq 0$ and $\mathcal{F}(t) \propto t^{-d/2}$ for $r = 0$. In particular, the algebraic decay observed for $r \neq 0$ in this weak-trap limit turns out to be the same as the one observed off criticality in the strong-trap limit, as also shown in panel (a) of Fig. 4 for $\Gamma^{(2)}(t) \propto \mathcal{F}(t)$, where the exponents of the decay of the curves for $\kappa \to 0$ and $\kappa \to \infty$ are equal.

In the weak-trap limit, the time $\omega_0^{-1}$ of the relaxation in the trap is much longer than the timescales appearing in the function $\mathcal{F}$, hence the integral in Eq. (45) which gives $C^{(2)}(t)$ in terms of $\mathcal{F}$ is eventually dominated by $u \simeq 0$, leading to

$$C^{(2)}(t) \simeq \frac{DT}{\gamma_0^2} t e^{-\omega_0 t} \int_0^\infty du \, \mathcal{F}(u) = \frac{DT}{\gamma_0^2} t e^{-\omega_0 t} \int \frac{d^d q}{(2\pi)^d} \frac{q^4 |V_q|^2}{\alpha_q(\alpha_q + Tq^2/\gamma_0)}. \tag{81}$$

Accordingly, upon increasing $t$, we expect $C^{(2)}(t)$ to increase linearly at short times with a proportionality coefficient that is not universal. This is clearly shown in the various plots of $C^{(2)}(t)$ reported in Fig. 2.

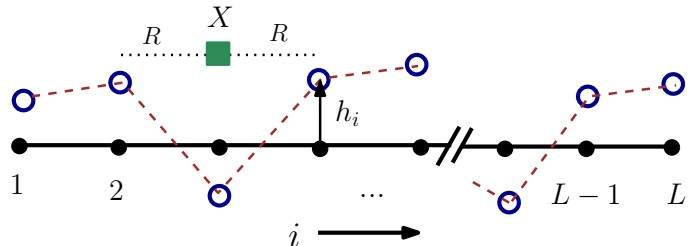

Figure 6: Schematic representation of the random walker coupled to a Rouse polymer model on a lattice, used for numerical simulations. The dashed red line indicates the polymer connecting $L$ monomers, represented as circles. The walker (green square) changes its position $X$ by jumping to one of its nearest neighbouring sites with a rate that depends on the displacement $h_i$ of the monomers within the range $X - R \leq i \leq X + R$.

## 5  Numerical simulations

In this section we provide numerical evidence to support the predictions formulated in the previous sections beyond the perturbation theory within which they have been derived. For the sake of simplicity, we focus on a dynamics occurring in one spatial dimension and consider a spatially discrete model for the colloid-field system.

The colloidal probe is modelled as a random walker moving on a periodic one-dimensional lattice of size $L$ and is coupled to a Rouse polymer chain, defined on the same lattice, which models the field, as sketched in Fig. 6. The modelling of the field as a Rouse chain is inspired by Ref. [28], where a similar chain is used for describing an Edward-Wilkinson interface. A continuous degree of freedom $h_i(t)$ is associated with each lattice site $i = 1, 2, \ldots L$ and it can be thought of as the displacement of the $i$-th monomer of the polymer, which represents the fluctuating field. The colloidal probe, with coordinate $X \in 1, 2, \cdots, L$ along the chain interacts with the field via the coupling potential $V(i, X)$. In the following, we consider the simple case $V(i, X) = \Theta(R - |i - X|)$, where $\Theta(x)$ is the unit step function, i.e., the colloid interacts with the field only within the interval $[X - R, X + R]$. In addition to the interaction with the polymer, the colloid is also trapped by a harmonic potential $\frac{\kappa}{2}(X - L/2)^2$, centered at $X = L/2$, also defined on the lattice. The Hamiltonian $H$ describing this coupled system is then given by

$$H = \sum_{i=1}^{L} \left[ \frac{1}{2}(\nabla h_i)^2 + \frac{r}{2} h_i^2 \right] + \frac{\kappa}{2} \left( X - \frac{L}{2} \right)^2 - \lambda \sum_{i=1}^{L} h_i V(i, X), \tag{82}$$

where $\nabla$ denotes the first-order discrete derivative on the lattice. The time evolution of the field follows the spatially discrete version of Eq. (6), i.e.,

$$\frac{dh_i}{dt} = D(r\Delta - \Delta^2)h_i - \lambda D \Delta V(i, X) - \nabla \eta_i(t), \tag{83}$$

where $\Delta$ denotes the discrete Laplacian operator, while $\eta_i$ are a set of $L$ independent Gaussian white noises. In defining the discrete operators $\nabla$, $\Delta$, and $\Delta^2$ we use the central difference scheme, i.e., for an arbitrary function $g_i$ with $i = 1, 2, \ldots$:

$$\begin{aligned}
\nabla g_i &= (g_{i+1} - g_{i-1})/2, \\
\Delta g_i &= g_{i+1} + g_{i-1} - 2g_i, \\
\Delta^2 g_i &= g_{i+2} + g_{i-2} + 6g_i - 4(g_{i+1} + g_{i-1}).
\end{aligned} \tag{84}$$

In the numerical simulations, the coupled first-order Langevin equations (83) and (84) are used to simulate the field dynamics.

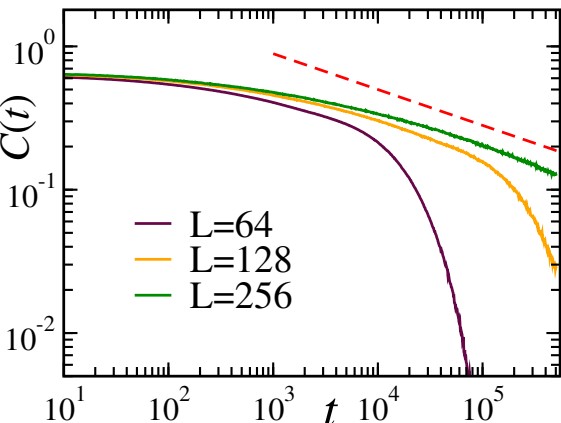

Figure 7: Numerical simulation of the colloid and the Rouse polymer model (see Fig. 6) in spatial dimension $d = 1$: Stationary auto-correlation function $C(t)$ of the position of the colloidal probe as a function of the time-difference $t$ at the bulk critical point $r = 0$ for various values of the lattice size $L$. The dashed line indicates the algebraic decay $t^{-1/4}$ predicted by the analytical perturbative calculation. Here the colloid radius $R$ is set to $R = 4$ and the coupling strength $\lambda = 0.5$, with all the remaining parameters set to one.

As anticipated above, the overdamped diffusive motion of the colloid is modelled as a random walker moving on the same lattice. From a position $X$, the walker jumps to one of its nearest neighbouring site $X' = X \pm 1$ with Metropolis rates $(T/\gamma_0) \min\{1, e^{-\Delta H/T}\}$ where $\Delta H = \kappa(X'^2 - X^2)/2 + \lambda \sum_i h_i[V(i, X) - V(i, X')]$ is the change in the energy $H$ in Eq. (82) due to the proposed jump $X \mapsto X'$.

The above dynamics ensures that the colloid-Rouse chain coupled system eventually relaxes to equilibrium with the Gibbs-Boltzmann measure corresponding to the Hamiltonian (82). We measure the temporal auto-correlation $C(t) = \langle X(0)X(t)\rangle$ of the colloid position $X$ in this equilibrium state. A plot of $C(t)$ as a function of time $t$ at criticality $r = 0$ for various values of lattice size $L$ is shown in Fig. 7. The expected algebraic decay $\propto t^{-1/4}$ of the long-time tail — predicted by Eq. (51) — becomes clearer as the system size $L$ increases and this agreement occurs for a generic choice of the system parameters. Accordingly, the data obtained from the numerical simulations, which is non-perturbative in nature, agrees very well with the theoretical prediction, providing a strong support to the perturbative approach presented in the previous sections.

## 6 Conclusions

This work presented a perturbative analytical study of the effective dynamics of a trapped overdamped Brownian particle which is linearly and reversibly coupled to a fluctuating Gaussian field with conserved dynamics (model B) and tunable spatial correlation length $\xi$.

In particular, in Sec. 3 we showed that the effective dynamics of the coordinate of the particle is determined by a non-Markovian Langevin equation (see Eq. (16)) characterised by a *non-linear* memory kernel determined by the dynamics of the field and by the interaction of the particle with the field. The effective noise in that equation turns out to be *colored and spatially correlated* in a way that is related to the memory kernel by the generalised fluctuation-dissipation relation discussed in Sec. 3.3. A possible *linear* approximation of this dynamics necessarily produces a linear memory kernel (see Sec. 3.2), which depends also on

the additional external forces and thus — contrary to the non-linear memory — is not solely determined by the interaction of the particle with the bath. As heuristically expected, this dependence becomes more pronounced upon making the dynamics of the field slower, i.e., upon approaching its critical point. A similar dependence was reported in numerical and experimental investigations of simple or viscoelastic fluids [8,9] as well as in theoretical studies of cartoon non-linear models of actual viscoelastic baths [9].

In Sec. 4 we determine the lowest-order perturbative correction to the equilibrium correlation function $C(t)$ of the particle position and to the associated power spectral density $S(\omega)$, due to the coupling to the field. We highlight the possible emergence of algebraic behaviours in the time-dependence at long times $t$ of $C(t)$ (see Eq. (53) and Fig. 2) and in the frequency-dependence at small frequencies $\omega$ of $S(\omega)$ (see Eq. (57) and Fig. 3), that are completely determined by the slow dynamics of the field which the particle is coupled to and that depend on whether the field is critical or not. The corresponding exponents turn out to be *universal*, as they are largely independent of the actual form of the interaction potential between the particle and the field. The effective (linear) memory kernel $\Gamma(t)$ (see Eq. (69) and Fig. 4) which can be inferred from the correlation function of the particle and the associated friction coefficient $\gamma$ (see Eq. (72) and Fig. 5) turn out to depend sensitively on the stiffness $\kappa$ of the external confining potential which the particle is subject to, especially when the field is poised at its critical point. In fact, correspondingly, fluctuations within the system are enhanced and the associated "fluctuation renormalisations" [29] of the linear coefficients are expected to be more relevant.

In Sec. 5 we probe the validity of the perturbative and analytical study of the dynamics of the particle via numerical simulations, confirming the algebraic decay of the correlations beyond perturbation theory (see Fig. 7).

In the present study we focused on the case of a Gaussian field with conserved dynamics, which provides a cartoon of a liquid medium and which is *generically* slow in the sense that the field correlation function displays an algebraic behaviour at long times also away from criticality. In the case of non-conserved dynamics (the so-called model A [12]), instead, the algebraic behaviour of correlations occurs only at criticality. Correspondingly, the non-critical algebraic decay of $C(t)$ in Eq. (53) is expected to be replaced by an exponential decay controlled by the field relaxation time $\tau_\phi \propto r^{-1}$ while the algebraic decay at criticality acquires a different exponent which can be determined based on a power-counting analysis. Some of these aspects have been recently studied in Ref. [30] together with the non-equilibrium relaxation of a particle which is released from an initial position away from the centre of the optical trap. Among the possible extensions of the present work, we mention considering a quadratic coupling of the particle to the field — which tends to move the probe towards the zeros of the field, — more general couplings [19], or the case of anisotropic particles having a polarity coupled to gradients of the fluctuating field. Similarly, it would be interesting to explore additional and experimentally observable consequences of the emergence of the effective non-linear equation of motion of the particle beyond the dependence of the linear coefficients on the external forces. In particular, as opposed to the case in which the evolution equation of the particle coordinate $X$ is linear, we expect that the statistics of suitably chosen observables should be non-Gaussian, in spite of the fact that the very stationary distribution of $X$ is Gaussian [31].

As a first step towards modelling colloidal particles in actual correlated fluids, instead, it would be important to consider more realistic models of dynamics, possibly including the case of non-Gaussian fields (which are expected to feature the emergence of critical exponents beyond mean-field), as well as of the interactions between the particle and the field, which usually take the form of boundary conditions for the latter. This would allow, inter alia, the investigation of the role of fluctuations in the dynamics of effective interactions among the particles immersed in the fluctuating medium [21,32–35], or a finer account of the effect of

viscoelasticity on the transition between two wells [36, 37].

## Acknowledgements

We thank Sergio Ciliberto, David S. Dean, Ignacio A. Martínez and Alberto Rosso for useful discussions.

**Funding information** AG acknowledges support from MIUR PRIN project "Coarse-grained description for non-equilibrium systems and transport phenomena (CO-NEST)" n. 201798CZL. UB acknowledges support from the Science and Engineering Research Board (SERB), India, under a Ramanujan Fellowship (Grant No. SB/S2/RJN-077/2018).

## A  Equilibrium distribution of the colloid position

In equilibrium, the distribution which characterises the fluctuations of the colloid position $X$ is given by

$$P(X) = \frac{e^{-H_X/T} \int \mathcal{D}\phi \; e^{-\mathcal{H}_\phi/T}}{\int \mathrm{d}X \int \mathcal{D}\phi \; e^{-\mathcal{H}/T}} , \tag{85}$$

where, for convenience, we have written $\mathcal{H} = \mathcal{H}_\phi + H_X$; $\mathcal{H}_\phi$ denotes the field-dependent part of the Hamiltonian in Eq. (1), including the interaction term between the colloid and the field, while $H_X = \kappa X^2/2$ denotes the contribution which solely depends on the colloid degree of freedom. Note that $\mathcal{H}_\phi$ depends on $X$ through the interaction potential $V$.

In order to evaluate the functional integration, it is useful to recast $\mathcal{H}_\phi$ in a bilinear form: using partial integration, we can formally express $\mathcal{H}_\phi = \int \mathrm{d}^d x \, (\phi \hat{A} \phi/2 - \lambda V \phi)$ where $\hat{A} = -\nabla^2 + r^2$. Because of the quadratic nature of $\mathcal{H}_\phi$, the functional integration can be exactly performed and yields

$$\int \mathcal{D}\phi \; e^{-\mathcal{H}_\phi/T} \propto \exp\left[ \frac{\lambda^2}{2T} \int \mathrm{d}^d y \; \mathrm{d}^d y' \; V(y) \hat{A}^{-1}(y - y') V(y') \right], \tag{86}$$

where $\hat{A}^{-1}$ denotes the inverse of the operator $\hat{A}$. Clearly, the right-hand side of Eq. (86) is independent of the colloid position $X$ and therefore the equilibrium probability distribution $P(X)$ of the coordinate $X$ of the particle in Eq. (85) is independent of the coupling to the field and is determined only by the trap.

## B  The fluctuation-dissipation relation

In this appendix, we show that a dynamics of the form

$$\gamma_0 \dot{X}(t) = \int_{-\infty}^{t} \mathrm{d}t' \, F(X(t) - X(t'), t - t') + \Xi(X(t), t), \tag{87}$$

where $\Xi(x, t)$ is a Gaussian noise with correlation

$$\langle \Xi_j(x, t) \Xi_l(x', t') \rangle = 2\gamma_0 T \delta_{jl} \delta(t - t') + T G_{jl}(x - x', t - t'), \tag{88}$$

is invariant under time reversal if the kernels $F(x,t)$ and $G(x,t)$ are related, for $t > 0$, as

$$\nabla_j F_l(x,t) = \partial_t G_{jl}(x,t). \tag{89}$$

Equation (87) corresponds to the effective evolution equation for the particle in interaction with the field, given by Eq. (16), but in the absence of the trap, i.e., for $\kappa = 0$. For the sake of simplicity we consider this case, as the argument presented below readily extends to $\kappa \neq 0$.

First, we note that if Eq. (89) is satisfied, we can introduce the potential

$$\Psi_j(x,t) = -\int_t^\infty dt' F_j(x,t'), \tag{90}$$

such that

$$F_j(x,t) = \partial_t \Psi_j(x,t), \tag{91}$$

$$G_{jl}(x,t) = \nabla_l \Psi_j(x,|t|). \tag{92}$$

Note that from Eq. (89), one infers that $\nabla_l F_j = \nabla_j F_l$ and $\nabla_l \Psi_j = \nabla_j \Psi_l$; also, by symmetry, $F(x,t)$ and $\Psi(x,t)$ should be odd functions of $x$. In order to simplify the notations, below we assume that both $F(x,t)$ and $\Psi(x,t)$ vanish for $t < 0$, while the integrals over time run over $\mathbb{R}$, unless specified otherwise.

First, we further simplify Eq. (87) by absorbing the delta correlation in Eq. (88) in the definition of $G(x,t)$ and the instantaneous friction $\gamma_0 \dot{X}(t)$ in the kernel $F$ via $F(X,t) \mapsto F(X,t) + \gamma_0 X \delta'_+(t)$. The dynamics now reads

$$-\int dt' F(X(t) - X(t'), t - t') = \Xi(X(t), t), \tag{93}$$

$$\langle \Xi_j(x,t) \Xi_l(x',t') \rangle = T G_{jl}(x - x', t - t'). \tag{94}$$

We introduce now the path-integral representation for this dynamics, following Ref. [13]. The corresponding Janssen-De Dominicis action is

$$S[X,P] = i \int dt\, dt'\, P_j(t) F_j(X(t) - X(t'), t - t')$$
$$+ \frac{T}{2} \int dt\, dt'\, P_j(t) P_l(t') G_{jl}(X(t) - X(t'), t - t'), \tag{95}$$

where $P(t)$ is the so-called response field (see, e.g., Sec. 4.1 in Ref. [38]).

In terms of the path-integral description of the process, we can now use the method presented in Ref. [39] to show that if the conditions expressed in Eqs. (91) and (92) are satisfied, the resulting (stationary) process is invariant under time-reversal, i.e., it is an equilibrium process. In particular, given a trajectory described by $\{X(t), P(t)\}$ we consider the corresponding time-reversed trajectory $\{\bar{X}(t), \bar{P}(t)\}$ with

$$\bar{X}(t) = X(-t), \tag{96}$$

$$\bar{P}(t) = P(-t) - \frac{i}{T} \dot{X}(-t). \tag{97}$$

In equilibrium, one should have $S[\bar{X}, \bar{P}] = S[X,P]$ and this is what we will check below. The

action of the reversed trajectory is thus

$$S[\bar{X},\bar{P}] = \frac{T}{2}\int dt\, dt'\, P_j(t)P_l(t')G_{jl}(X(t)-X(t'),t-t') \tag{98}$$

$$+ i\int dt\, dt'\, P_j(t)\big[F_j(X(t)-X(t'),t'-t) - \dot{X}_l(t')G_{jl}(X(t)-X(t'),t-t')\big]$$

$$+ \frac{1}{T}\int dt\, dt'\, \dot{X}_j(t)\Big[F_j(X(t)-X(t'),t'-t) - \frac{1}{2}\dot{X}_l(t')G_{jl}(X(t)-X(t'),t-t')\Big].$$

The first term coincides with the second one in $S[X,P]$ in Eq. (95). The second and the last term in $S[X,P]$ can now be rewritten by assuming that Eqs. (91) and (92) hold. For the second term we have

$$\int dt\, dt'\, P_j(t)\big[F_j(X(t)-X(t'),t'-t) - \dot{X}_l(t')G_{jl}(X(t)-X(t'),t-t')\big]$$

$$= \int dt\, dt'\, P_j(t)\big[\partial_t\Psi_j(X(t)-X(t'),t'-t) - \dot{X}_l(t')\nabla_l\Psi_j(X(t)-X(t'),|t-t'|)\big] \tag{99}$$

$$= \int_{t'>t} dt\, dt'\, P_j(t)\big[\partial_t\Psi_j(X(t)-X(t'),t'-t) - \dot{X}_l(t')\nabla_l\Psi_j(X(t)-X(t'),t'-t)\big]$$

$$+ \int_{t'<t} dt\, dt'\, P_j(t)\big[-\dot{X}_l(t')\nabla_l\Psi_j(X(t)-X(t'),t-t')\big] \tag{100}$$

$$= I_{t'>t} + I_{t'<t}. \tag{101}$$

The first integral vanishes because

$$I_{t'>t} = \int_{t'>t} dt\, dt'\, P_j(t)\frac{d}{dt'}\big[\Psi_j(X(t)-X(t'),t'-t)\big] = 0. \tag{102}$$

We have used that $\Psi_j(x,t\to\infty)=0$ and $\Psi_j(0,0)=0$, given that $\Psi_j(x,t)$ is an odd function of $x$. We use the same trick to integrate by parts in the second integral in Eq. (101) (the boundary terms cancel):

$$I_{t'<t} = \int_{t'<t} dt\, dt'\, P_j(t)\partial_t\Psi_j(X(t)-X(t'),t-t') \tag{103}$$

$$= \int dt\, dt'\, P_j(t)F_j(X(t)-X(t'),t-t'). \tag{104}$$

This term coincides with the first one in the action $S[X,P]$ of the forward path in Eq. (95) and thus, in order to prove that $S[\bar{X},\bar{P}] = S[X,P]$, we have to show that the last term in Eq. (98) actually vanishes.

This contribution can be calculated as above, except that we start by using the parity of $G(x,t)$ to remove the factor $1/2$ and to restrict the integral to $t'>t$:

$$\int dt\, dt'\, \dot{X}_j(t)\Big[F_j(X(t)-X(t'),t'-t) - \frac{1}{2}\dot{X}_l(t')G_{jl}(X(t)-X(t'),t-t')\Big]$$

$$= \int_{t'>t} dt\, dt'\, \dot{X}_j(t)\big[F_j(X(t)-X(t'),t'-t) - \dot{X}_l(t')G_{jl}(X(t)-X(t'),t-t')\big] \tag{105}$$

$$= \int_{t'>t} dt\, dt'\, \dot{X}_j(t)\big[\partial_t\Psi_j(X(t)-X(t'),t'-t) - \dot{X}_l(t')\nabla_l\Psi_j(X(t)-X(t'),t'-t)\big] \tag{106}$$

$$= \int_{t'>t} dt\, dt'\, \dot{X}_j(t)\frac{d}{dt'}\big[\Psi_j(X(t)-X(t'),t'-t)\big] = 0, \tag{107}$$

which completes the proof.

# C   Correlation function of the position of the particle

In this appendix we provide detailed derivation of the $\mathcal{O}(\lambda^2)$ correction $C^{(2)}(t)$ to the stationary state autocorrelation of the colloid, i.e., we derive Eq. (45). In particular, in Sec. C.1 we first derive some identities concerning relevant two- and three-time correlation functions of the Ornstein-Uhlenbeck process which are needed in Sec. C.2 in order to calculate $C^{(2)}(t)$ perturbatively. In Sec. C.3, instead, we derive the expression of the long-time behaviour of the correction $C^{(2)}(t)$ and relate it to $\mathcal{F}(t)$.

## C.1   Correlations in the Ornstein-Uhlenbeck process

Let us first consider the two-time correlation which we will need, c.f., in Eq. (120):

$$\left\langle e^{-iq\cdot X^{(0)}(s)} e^{iq\cdot X^{(0)}(s')} \right\rangle = \prod_{j=1}^{d} \left\langle e^{-iq_j X_j^{(0)}(s_1)} e^{iq_j X_j^{(0)}(s_2)} \right\rangle, \tag{108}$$

where the statistical average is over the probe trajectories in the stationary state for $\lambda = 0$. In this case, each component of the probe position undergoes an independent Ornstein-Uhlenbeck (OU) process following Eq. (30). Consequently, it suffices to calculate

$$\left\langle e^{-\alpha x(s_1)} e^{\alpha x(s_2)} \right\rangle = \lim_{t_0 \to -\infty} \int dx_1 \, dx_2 \, e^{-\alpha x_1} e^{\alpha x_2} P(x_2, s_2 | x_1, s_1) P(x_1, s_1 | x_0, t_0), \tag{109}$$

where we assumed $s_2 > s_1$ (the opposite case can be obtained with $\alpha \to -\alpha$) and where $P(x, t|y, s)$ denotes the probability that an OU particle, starting from position $y$ at time $s$ will reach position $x$ at a later time $t$. The Gaussian white noise $\zeta(t)$ driving the dynamics in Eq. (30) ensures that $P(x, t|y, s)$ is Gaussian and given by

$$P(x, t|y, s) = \frac{1}{\sqrt{4\pi\gamma(t, s)}} \exp\left\{ -\frac{\left[x - y e^{-\omega_0(t-s)}\right]^2}{4\gamma(t, s)} \right\} \text{ with } \gamma(t, s) = \frac{T}{2\kappa}[1 - e^{-2\omega_0(t-s)}]. \tag{110}$$

This expression can now be used in Eq. (109) in order to calculate its l.h.s. via a Gaussian integration over $x_1$ and $x_2$. As we are interested in the stationary state only, we take $t_0 \to -\infty$, which leads to

$$\left\langle e^{-\alpha x(s_1)} e^{\alpha x(s_2)} \right\rangle = \exp\left\{ -\frac{\alpha^2 T}{\kappa} \left[1 - e^{-\omega_0(s_2-s_1)}\right] \right\}, \tag{111}$$

where we assumed $s_2 > s_1$. Finally, substituting $\alpha = iq_j$, and taking the product over $j$ [see Eq. (108)], we have

$$\left\langle e^{-iq\cdot X^{(0)}(s)} e^{iq\cdot X^{(0)}(s')} \right\rangle = \exp\left\{ -\frac{q^2 T}{\kappa} \left[1 - e^{-\omega_0|s-s'|}\right] \right\}. \tag{112}$$

In Eq. (124) below we will need an analytic expression for three-time correlation of the form

$$\left\langle e^{iq\cdot X^{(0)}(s_1)} e^{-iq\cdot X^{(0)}(s_2)} X_j^{(0)}(s_3) \right\rangle, \tag{113}$$

which are also computed following the same procedure as above. In particular, in the stationary state, one eventually finds

$$\left\langle e^{iq\cdot X^{(0)}(s_1)} e^{-iq\cdot X^{(0)}(s_2)} X_j^{(0)}(s_3) \right\rangle = \frac{iq_j T}{\kappa} \left[ e^{-\omega_0|s_3-s_1|} - e^{-\omega_0|s_3-s_2|} \right]$$
$$\times \exp\left\{ -\frac{q^2 T}{\kappa} \left[1 - e^{-\omega_0|s_2-s_1|}\right] \right\}. \tag{114}$$

## C.2 Perturbative correction

We start with the perturbative solutions for $X^{(n)}(t)$ in Eq. (33). Substituting the explicit expressions for

$$f_j^{(0)}(t) \equiv f_j(t)|_{\lambda=0} = -i \int \frac{d^d q}{(2\pi)^d} q_j V_{-q} \phi_q^{(0)}(t) e^{-iq \cdot X^{(0)}(t)}, \tag{115}$$

$$f_j^{(1)}(t) \equiv \frac{df_j(t)}{d\lambda}|_{\lambda=0} = -i \int \frac{d^d q}{(2\pi)^d} q_j V_{-q} \left[ \phi_q^{(1)}(t) - iq \cdot X^{(1)}(t) \phi_q^{(0)}(t) \right] e^{-iq \cdot X^{(0)}(t)}, \tag{116}$$

we get

$$X_j^{(1)}(t) = -i\gamma_0^{-1} \int_{-\infty}^t ds\, e^{-\omega_0(t-s)} \int \frac{d^d q}{(2\pi)^d} q_j V_{-q} \phi_q^{(0)}(s) e^{-iq \cdot X^{(0)}(s)}, \tag{117}$$

$$X_j^{(2)}(t) = -i\gamma_0^{-1} \int_{-\infty}^t ds\, e^{-\omega_0(t-s)} \int \frac{d^d q}{(2\pi)^d} q_j V_{-q} e^{-iq \cdot X^{(0)}(s)} \int_{-\infty}^s ds'$$

$$\times \left[ Dq^2 V_q e^{-\alpha_q(s-s')} e^{iq \cdot X^{(0)}(s')} \right.$$

$$\left. - \gamma_0^{-1} e^{-\omega_0(s-s')} \int \frac{d^d q'}{(2\pi)^d} q \cdot q' V_{-q'} \phi_{q'}^{(0)}(s') \phi_q^{(0)}(s) e^{-iq' \cdot X^{(0)}(s')} \right]. \tag{118}$$

Using the above equations we can compute the three different terms appearing in $C^{(2)}(t)$ [see Eq. (44)]. Let us first compute

$$C_1^{(2)}(t_2 - t_1) \equiv \langle X_j^{(1)}(t_1) X_j^{(1)}(t_2) \rangle$$

$$= -\gamma_0^{-2} \int_{-\infty}^{t_1} ds\, e^{-\omega_0(t_1-s)} \int_{-\infty}^{t_2} ds'\, e^{-\omega_0(t_2-s')} \int \frac{d^d q}{(2\pi)^d} q_j V_{-q}$$

$$\times \int \frac{d^d q'}{(2\pi)^d} q'_j V_{-q'} \left\langle \phi_q^{(0)}(s) \phi_{q'}^{(0)}(s') \right\rangle \left\langle e^{-iq \cdot X^{(0)}(s)} e^{-iq' \cdot X^{(0)}(s')} \right\rangle, \tag{119}$$

where $\langle \cdot \rangle$ denotes statistical averages over the decoupled Gaussian field and probe trajectories. Using Eq. (39) for the free Gaussian field correlation and performing the $q'$ integral, we get

$$C_1^{(2)}(t_2 - t_1) = \frac{DT}{\gamma_0^2} \int_{-\infty}^{t_1} ds\, e^{-\omega_0(t_1-s)} \int_{-\infty}^{t_2} ds'\, e^{-\omega_0(t_2-s')} \int \frac{d^d q}{(2\pi)^d} \frac{q_j^2 q^2}{\alpha_q} |V_q|^2$$

$$\times \left\langle e^{-iq \cdot X^{(0)}(s)} e^{iq \cdot X^{(0)}(s')} \right\rangle. \tag{120}$$

The auto-correlation of the probe particle in the last expression has been calculated above in Eq. (112) and, after a change of variables $u = t_1 - s'$ and $v = t_1 - s$, we get

$$C_1^{(2)}(t) = \frac{DT}{\gamma_0^2} e^{-\omega_0 t} \int_0^\infty dv\, e^{-\omega_0 v} \int_{-t}^\infty du\, e^{-\omega_0 u} \mathcal{F}_j(|u-v|), \tag{121}$$

where $t = t_2 - t_1$ and we have introduced

$$\mathcal{F}_j(z) = \int \frac{d^d q}{(2\pi)^d} \frac{q_j^2 q^2}{\alpha_q} |V_q|^2 \exp\left[ -\alpha_q z - \frac{q^2 T}{\kappa}(1 - e^{-\omega_0 z}) \right]. \tag{122}$$

Performing the $v$-integral, we arrive at a simpler expression,

$$C_1^{(2)}(t) = \frac{DT}{2\kappa\gamma_0} e^{-\omega_0 t} \left[ \int_0^\infty du\, e^{-\omega_0 u} \mathcal{F}_j(u) + \int_0^t du\, e^{\omega_0 u} \mathcal{F}_j(u) + e^{2\omega_0 t} \int_t^\infty du\, e^{-\omega_0 u} \mathcal{F}_j(u) \right]. \tag{123}$$

Next, we calculate

$$
\begin{aligned}
C_2^{(2)}(t_1, t_2) &= \langle X_j^{(0)}(t_1) X_j^{(2)}(t_2) \rangle \\
&= -i\gamma_0^{-1} D \int_{-\infty}^{t_2} ds \, e^{-\omega_0(t_2-s)} \int \frac{d^d q}{(2\pi)^d} \frac{q_j q^2}{\alpha_q} |V_q|^2 \int_{-\infty}^{s} ds' e^{-\alpha_q(s-s')} \\
&\quad \times \left[ \alpha_q + \gamma_0^{-1} T q^2 e^{-\omega_0(s-s')} \right] \left\langle e^{iq \cdot X^{(0)}(s')} e^{-iq \cdot X^{(0)}(s)} X_j^{(0)}(t_1) \right\rangle,
\end{aligned}
\tag{124}
$$

where we used Eq. (39). The three-time correlation for the probe trajectory in the previous expression was determined in Eq. (114) and, after assuming $t_2 > t_1$, the variable transformations $u = s - s'$ and $v = t_1 - s$ lead to

$$
C_2^{(2)}(t) = -\frac{DT}{\kappa \gamma_0} e^{-\omega_0 t} \int_{-t}^{\infty} dv \, e^{-\omega_0 v} \int_0^{\infty} du \, \frac{d\mathcal{F}_j}{du} (e^{-\omega_0|u+v|} - e^{-\omega_0|v|}).
\tag{125}
$$

Performing a partial integration over $u$ and the $v$-integral, we get

$$
C_2^{(2)}(t) = \frac{DT}{\kappa \gamma_0} e^{-\omega_0 t} \left[ \int_0^{t} du \, e^{\omega_0 u} (2\omega_0(t-u) - 1) \mathcal{F}_j(u) - e^{2\omega_0 t} \int_t^{\infty} du \, e^{-\omega_0 u} \mathcal{F}_j(u) \right].
\tag{126}
$$

Lastly, we need to determine $C_3^{(2)}(t_1, t_2) = \langle X_j^{(2)}(t_1) X_j^{(0)}(t_2) \rangle$ which is obtained from the expression of $C_2^{(2)}(t_1, t_2)$ in Eq. (124) by exchanging $t_1$ with $t_2$. After assuming $t_2 > t_1$ and using the three-time correlation of the probe trajectory determined in Eq. (114) and the change of variables $u = s - s'$ and then $v = t_1 - s$, we arrive at

$$
C_3^{(2)}(t_1, t_2) = -\frac{DT}{\kappa \gamma_0} e^{-\omega_0 t} \int_0^{\infty} dv \, e^{-2\omega_0 v} \int_0^{\infty} du \, \frac{d\mathcal{F}_j}{du} (e^{-\omega_0 u} - 1).
\tag{127}
$$

Performing a partial integration over $u$ and the $v$-integral, we have

$$
C_3^{(2)}(t) = -\frac{DT}{2\kappa \gamma_0} e^{-\omega_0 t} \int_0^{\infty} du \, e^{-\omega_0 u} \mathcal{F}_j(u).
\tag{128}
$$

Finally, adding Eqs. (123), (126), and (128), and summing over $j$, we have a simple expression for the second-order correction to the auto-correlation, i.e., Eq. (45) in which $\mathcal{F}(u) = \sum_{j=1}^{d} \mathcal{F}_j(u)$, i.e., taking into account Eq. (122), $\mathcal{F}$ is given by Eq. (46).

## C.3 Long-time behaviour

In order to determine the algebraic behaviour of $C^{(2)}(t)$ at long times, we consider $t \gg \omega_0^{-1}$ (or, formally, $\omega_0 \to \infty$ with fixed $t$) and note that the factor $e^{-\omega_0(t-u)}(t-u)$ in the integrand of Eq. (45) takes its maximal value $\propto \omega_0^{-1}$ at $u \simeq t - \omega_0^{-1}$, quickly vanishing away from it. Accordingly, upon increasing $\omega_0$, this factor provides an approximation of $\omega_0^{-2} \delta_+(t-u)$. As a consequence, in the limit $t \gg \omega_0^{-1}$, Eq. (45) renders

$$
C^{(2)}(t) = \frac{DT}{\gamma_0^2} \int_0^{t} du \, e^{-\omega_0(t-u)}(t-u) \mathcal{F}(u) \simeq \frac{DT}{\gamma_0^2 \omega_0^2} \mathcal{F}(t),
\tag{129}
$$

i.e., Eq. (47) after taking into account Eq. (14).

# D  Power spectral density and its asymptotic behavior

In this Appendix we determine the power spectral density $S(\omega)$, i.e., the Fourier transform of $C(t)$ and discuss its asymptotic behaviours.

## D.1  General expression

The correction $S^{(2)}(\omega)$ of $\mathcal{O}(\lambda^2)$ to the power spectral density $S(\omega)$ is obtained by taking the Fourier transform of Eq. (45) w.r.t. time $t$, i.e.,

$$S^{(2)}(\omega) = \int_0^\infty dt\, (e^{i\omega t} + e^{-i\omega t}) C^{(2)}(t) = \frac{DT}{\gamma_0^2}[Z(\omega) + Z^*(\omega)], \qquad (130)$$

where we introduced

$$
\begin{aligned}
Z(\omega) &= \int_0^\infty dt\, e^{i\omega t} \int_0^t du\, e^{-\omega_0(t-u)}(t-u)\mathcal{F}(u) \\
&= \frac{1}{(\omega_0 - i\omega)^2} \int_0^\infty du\, e^{i\omega u}\mathcal{F}(u),
\end{aligned}
\qquad (131)
$$

where, in the last line, we used the fact that $\int_0^\infty dt \int_0^t du = \int_0^\infty du \int_u^\infty dt$. Inserting this expression in Eq. (130), with some straightforward algebra, one readily derives Eq. (55).

## D.2  Asymptotic behaviors

Here we determine the asymptotic behaviours of $S(\omega)$ for $\omega \to 0$ and then for $\omega \to \infty$. As the behavior of $S^{(0)}(\omega)$ can be easily derived from Eq. (15), we focus below on the contribution $S^{(2)}(\omega)$ due to the coupling to the field. In particular, the leading asymptotic behavior of $S^{(2)}(\omega)$ in the limit $\omega \to 0$ is obtained from Eq. (55):

$$S^{(2)}(\omega \to 0) \simeq \frac{2DT}{\kappa^2} \int_0^\infty du\, \mathcal{F}(u)\cos(\omega u), \qquad (132)$$

with $\mathcal{F}(u)$ given in Eq. (46) and displays, for large $t$, the scaling behaviour highlighted in Eqs. (49) and (50). Assuming that the interaction $V_q$ regularises the possible divergence of the integral defining $\mathcal{F}$, such that $\mathcal{F}(0)$ is finite, we need to consider the integrand in Eq. (132) at large $u$, for which we have that $\mathcal{F}(u) \propto u^{-(1+d/2)}$ for $r > 0$ and $\mathcal{F}(u) \propto u^{-d/4}$ for $r = 0$. Accordingly, for $r > 0$ or $d \geq 4$, $\int_0^\infty du\, \mathcal{F}(u)$ is finite and

$$S^{(2)}(\omega = 0) = \frac{2DT}{\kappa^2} \int_0^\infty du\, \mathcal{F}(u). \qquad (133)$$

For the purpose of understanding the dependence of the effective friction $\gamma$ (see, c.f., Eq. (72)) on the trap strength $\kappa$, we investigate here the behaviour of the integral in Eq. (133) in the two formal limits $\kappa \to 0$ and $\kappa \to \infty$, corresponding to weak and strong trapping, respectively. Taking into account that $\omega_0 \propto \kappa$ (see Eq. (14)), these limits can be taken in the integrand of Eq. (46) and the remaining integrals yield

$$S^{(2)}(\omega = 0; \kappa \to 0) = \frac{2T}{D\kappa^2} \int \frac{d^d q}{(2\pi)^d} \frac{|V_q|^2}{(q^2 + r)[q^2 + r + T/(D\gamma_0)]}, \qquad (134)$$

and

$$S^{(2)}(\omega = 0; \kappa \to \infty) = \frac{2T}{D\kappa^2} \int \frac{\mathrm{d}^d q}{(2\pi)^d} \frac{|V_q|^2}{(q^2 + r)^2}. \tag{135}$$

In particular, upon approaching the critical point with $r \to 0$, these expressions might display a singular dependence on $r$, which is essentially determined by the behaviour of the corresponding integrands for $q \to 0$. In fact, one finds that

$$S^{(2)}(\omega = 0; \kappa \to 0; r \to 0) \sim \frac{2\gamma_0}{\kappa^2} |V_0|^2 r^{-1+d/2} \frac{\Gamma(1 - d/2)}{(4\pi)^{d/2}} \quad \text{for} \quad d < 2, \tag{136}$$

and

$$S^{(2)}(\omega = 0; \kappa \to \infty; r \to 0) \sim \frac{2T}{D\kappa^2} |V_0|^2 r^{-2+d/2} \frac{\Gamma(2 - d/2)}{(4\pi)^{d/2}} \quad \text{for} \quad d < 4, \tag{137}$$

while they tend to finite values otherwise. On the other hand, when $r = 0$ and $d < 4$, the integral in Eq. (133) does not converge but we can use for $\mathcal{F}(u)$ in Eq. (132) the approximate expression at long times given by Eq. (48) with $\alpha_q = Dq^4$. After some changes of variables, one finds

$$S^{(2)}(\omega \to 0; r = 0) \sim \frac{2T}{\kappa^2} \frac{V_0^2}{D^{d/4}} \omega^{-1+d/4} \frac{\Omega_d}{(2\pi)^d} \int_0^\infty \mathrm{d}x \, x^{d-1} \int_0^\infty \mathrm{d}v \, e^{-x^4 v} \cos v. \tag{138}$$

The remaining integrals can be done analytically and yield the finite constant $\pi/[8\cos(\pi d/8)]$ and therefore

$$S^{(2)}(\omega \to 0; r = 0) \sim \frac{\pi \Omega_d}{4(2\pi)^d \cos(\pi d/8)} \frac{T V_0^2}{\kappa^2 D^{d/4}} \omega^{-1+d/4}, \tag{139}$$

which is consistent with Eq. (57) in the main text, obtained by using the late-time behaviour of the probe auto-correlation.

In order to determine the behaviour of $S^{(2)}(\omega)$ for $\omega \to \infty$ (physically understood as taking $\omega$ larger than any other frequency scale in the problem) we focus on the expression of $S^{(2)}(\omega)$ in Eqs. (130) and (131) and therefore consider the asymptotic behaviour of

$$\int_0^\infty \mathrm{d}u \, e^{i\omega u} \mathcal{F}(u) = i\frac{\mathcal{F}(0)}{\omega} - \frac{\mathcal{F}'(0)}{\omega^2} + \mathcal{O}(\omega^{-3}). \tag{140}$$

This expansion is obtained by using the Riemann-Lebesgue lemma and successive integrations by parts, using that $\mathcal{F}(u)$, $\mathcal{F}'(u)$ and $\mathcal{F}''(u)$ are integrable. Inserting this expansion in Eqs. (131) and (130), one readily finds

$$S^{(2)}(\omega \to \infty) = -\frac{2DT}{\gamma_0^2} \frac{2\omega_0 \mathcal{F}(0) - \mathcal{F}'(0)}{\omega^4} + \mathcal{O}(\omega^{-5}), \tag{141}$$

where the numerator of the leading behaviour $\propto \omega^{-4}$ is a positive quantity given by

$$2\omega_0 \mathcal{F}(0) - \mathcal{F}'(0) = \int \frac{\mathrm{d}^d q}{(2\pi)^d} q^4 |V_q|^2 \left(1 + \frac{2\kappa + Tq^2}{\gamma_0 \alpha_q}\right). \tag{142}$$

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
