# Peer review of "Dynamics of a colloidal particle coupled to a Gaussian field: from a confinement-dependent to a non-linear memory"

_SciPost Physics, doi:SciPost Phys. 13, 078 (2022)_

## Round 1 · Author Response

Reply to Report 1

Referee's Comment: The authors provide a detailed theoretical study of the dynamics of a colloidal particle in a trap driven through a complex, phase-separating fluid. [....] Altogether, the authors present an interesting and careful study of a problem of current interest, that is, the non-Markovian response of a particle in a viscoelastic medium which cannot be described via simple (linear) kernels often used in the literature. They clarify the implicit dependencies on trap stiffness etc. found in earlier experiments and simulations, which is certainly a very relevant result. The here envisioned phase-separating medium can be studied experimentally, and I really like to dependency of the results on the underlying correlation length.

Reply: We thank the Referee for the detailed and constructive report, and for appreciating the scientific merit of our work. Below we provide an itemized reply to all points. The part of the manuscript which have been revised are highlighted in blue in the new version.

Referee's Comment: However, I am not convinced that the study contains sufficiently innovative pieces which would justify a publication in SciPost. My main concern is that the model equations (2) and (3) have already been used in the literature (see e.g. [21,22]), so the very idea of coupling the colloid to a density field to describe its dynamics is not new. Also the non-Markovian equation (16) leading to the non-trivial memory kernel has been considered previously (see Eq. (33) in [21]), and a (technically slightly different) perturbation analysis in terms of $\lambda$ has been proposed earlier in [21,22], as noted by the authors themselves. So in my impression, the main new approach here is the analysis of the role of the underlying correlation length... (?) In any case, the authors need to better clarify these issues in the manuscript. In the present form, the manuscript seems more suitable for a specialized journal.

Reply: As the Referee points out, the model of a probe particle linearly coupled to a Gaussian field has already been considered in the literature in some physical configurations which are: - Ref. [13], free particle, i.e., no external potential: calculation of the long-time diffusion coefficient; - Ref.[21], particle dragged by a constant force acting on it: calculation of the effective mobility and diffusion coefficient; - Ref.[22], particle dragged by a harmonic trap moving at a constant speed: calculation of the mean and variance of the position. Here, we consider a probe held in a static harmonic trap. Although it can formally be considered a special case of the model studied in Ref.[22], we focus on completely different observables, not analysed before. In particular, we consider the two-time correlation function of the position of the probe particle, with a different aim than the previous studies. Several effective parameters can be actually deduced from this quantity, for instance the long-time diffusion coefficient investigated in Ref.[13] (but we do not discuss this connection in the current manuscript). We decided to focus on this observable because it is used in actual experiments and numerical simulations for inferring the effective memory kernel of the medium, as we also do here in order to explain how its dependence on the trapping potential emerges. In addition, as noted by the Referee, we analyse here in detail the algebraic decay of this two-time correlations as functions of the correlation length of the field. In this respect, although the model has been already used in the literature, our manuscript focuses on issues which were not previously investigated in spite of the fact that they have important consequences for the interpretation of numerical and experimental data. In order to clarify this point, the sentence ``The model described above and variations thereof have been used in the literature in order to investigate theoretically the dynamics of freely diffusing or dragged particles [13,21,22], in the bulk or under spatial confinement [14] as well as the field-mediated interactions among particles and their phase behavior [23--25]." which originally appeared after Eq.(5) has been moved to the introduction, and modified as highlighted in blue.

Referee's Comment: Equation (1): I understand that an effective Hamiltonian quadratic in the field is convenient for the later analysis, since the equations of motion are linear in the field. However, why is this ansatz still justified for a fluid close to the critical point - shouldn't one go (at least) to fourth order to ensure stability? Or do these higher-order terms cancel out anyway in the later analysis?

Reply: In this manuscript we consider only the case in which the parameter $r$ controlling the distance form the critical point is positive, i.e., $r \ge 0$, corresponding to the fluid medium being in its homogeneous (mixed) phase or at its critical point. In these cases the thermodynamics and dynamics is stable without the need of introducing the fourth-order term mentioned by the Referee. Clearly, the latter is necessary in order to be able to provide a quantitative description of the critical behavior of an actual demixing fluid, which also requires considering a more complicated model of dynamics. In order to avoid these complications --- which will probably obscure the main message of the manuscript --- we decided to focus on a case in which at least the effective dynamics (Sec.3) can be determined exactly, which requires the dynamics of the field to be linear and thus the effective field Hamiltonian to be quadratic.

Referee's Comment: It seems to be important to have a rotationally invariant potential $V(x)$, see, e.g., Eq. (23) below. The authors should comment on the physical implications of this restriction.

Reply: We use an isotropic potential $V(x)$ in order to obtain simpler and more tractable expressions. However, there is no technical restriction on the shape of the potential and all the calculations can be generalized to that case. For instance, the effective memory kernel [appearing, e.g., in Eq.(23)] is a second-rank tensor in general, and it becomes proportional to the identity matrix when $V(x)$ is isotropic. In generalizing the problem to anisotropic potentials (representing, e.g., Janus-like particles characterized by some sort of dipole), one might also need to introduce the dynamics of additional degrees of freedoms, such as the ``orientation'' of such as anisotropic particle, which would make the dynamics of the problem certainly richer but beyond the scope of the present work. In order to clarify this point we have modified and expanded the text as highlighted in blue in the paragraph which follows Eq. (1).

Referee's Comment: Beginning of Sec. 4.1: I do not really understand what the authors mean by "equations .... are made non-linear by their coupling lambda". After all, the equations ARE linear in lambda ....

Reply: We actually mean that the equation for the probe position $X(t)$ becomes non-linear upon introducing the coupling $\propto \lambda$ to the field. For $\lambda =0$ the dynamics of the probe is clearly linear (see Eq.(7) with $\lambda =0$) and corresponds to a standard Ornstein-Uhlenbeck process. The position $X(t)$ of the probe, however, enters non-linearly in the dynamics of the field [Eq. (6)], which, in turn, affects the dynamics of the probe, and therefore the effective dynamics of the probe obtained after integrating out the field is non-linear, as clearly shown by Eqs.(16) and (17). In order to clarify this fact we have rephrased the sentence in Sec.4.1, as highlighted in blue in the revised version.

Reply to Report 2

Referee's Comment: The authors investigate the equilibrium dynamics of a colloidal particle embedded in a fluctuating medium described by a Gaussian field to which the particle is linearly coupled (the coupling amplitude is a nonlinear function of the colloid's position). [....] Overall this work addresses tracer diffusion in a complex medium, which is a notoriously difficult problem. The model is carefully defined and the calculations are presented in a very well-organized fashion. Numerical simulations are also presented with a view to delineating the domain of validity of the perturbation expansion. Actually, my only critical comments are mostly concerned with discussion and presentation issues (in the introduction). If the authors addressed these comments, my opinion would be that the submission could be accepted.

Reply: We thank the Referee for the detailed, constructive, and supportive report. Below we provide an itemized reply to all its points. The part of the manuscript which have been changed for addressing these issues are highlighted in blue in the revised version.

Referee's Comment: 1) In the model, the coupling of the particle to the medium appears to be linear in the field (which of course allows for integrating out of the $\phi$ modes). A linear coupling affects the deterministic field profile, but leaves the field fluctuations unaffected (and there is no corresponding free energy contribution). The peculiar choice of a linear coupling (in $\phi$) deserves a deeper discussion (this is also discussed in the conclusion of [13], and likely the author common to [13] and to the present submission has an opinion on whether the linear nature of the coupling to the field matters or not). I am sure that the authors can do a little more than the one sentence that addresses this issue in the conclusion section.

Reply: We thank the Referee for pointing out that further comments are necessary on this issue. In fact, in order to be able to determine the effective equation of motion of the probe coordinate $X$ in a non-perturbative fashion, one should be able to determine exactly the effect of the presence of the probe on the field configurations and, in case, on its fluctuations. In turn, this requires that the equation of motion for the field $\phi$ for a given position $X$ of the probe is linear and therefore that the coupling of the particle to the medium in the Hamiltonian is at most quadratic. However, as discussed in V. D\'emery, Phys. Rev. E {\bf 87}, 052105 (2013), the case of a quadratic coupling, in general, does not allow for a non-perturbative solution and therefore the linear coupling is the only one which yields an exact (i.e., non-perturbative) effective dynamics of the probe. Having such an equation at our disposal is important because the calculation is much simpler and it shows neatly the effect that we wish to describe, i.e., an effective dynamics which depends on the external forces. Moreover, a linear coupling implies that the presence of the particle breaks the symmetry $\phi \leftrightarrow -\phi$ characterizing the fluctuations of the unperturbed medium. In terms of the order parameter $\phi$, this means that the particle preferentially adsorb one of the two competing phases of the system, which is generically the case when colloidal particles are immersed in binary liquid mixtures, see, e.g., Refs.[16,32,33] of the manuscript. In order to spell out these facts we have added a comment at the end of the first paragraph in Sec.~2.1, where we introduce the coupling, see the text in blue in the revised version.

Referee's Comment: 2) In their introduction, the authors comment on the $X$ dependence of $\Gamma$: I understand that in an introduction one has to stage ones' results, but perhaps some of the "surprise" could be toned down ("annoying feature", "undesired", really ?). I don't think that there is any prior expectation that $\Gamma$ should be independent of $X$. In the standard Mori-Zwanzig projection formalism (repeated in [9]), such a dependence is actually a natural feature. It is only because additional hypotheses (in terms of separation of energy, length and time scales) are fulfilled that eventually $\Gamma$ can become independent of $X$ (in short, the mobility in a thermostat can depend on the thermostated degrees of freedom: a textbook example is that of a freely diffusing particle in the vicinity of a wall, where hydrodynamics makes the mobililty position dependent). I have the superficial impression that these arguments/comments are already clearly stated in [8,9] and it wouldn't hurt (in my opinion) for the clarity of the manuscript to plainly repeat them and then to insist on the core of the manuscript (how to describe the motion when the medium is a Gaussian field). In fact, and in my opinion, the manuscript is interesting in that it defines a model that is simple enough to be attacked by analytical means, but complex enough to the point of displaying such nontrivial features such as a position dependent mobility.

Reply: Although we agree with the Referee that, in general, there is no theoretical reason why $\Gamma$ should not depend on $X$, for practical purposes this is often assumed to be the case, with possibly a velocity-dependent friction coefficient. This assumption is usually and practically done when extracting the properties of a fluid via microrheology experiments or via numerical simulations and this is the reason why it comes often as an annoying feature that the extracted coefficients are not solely determined by the coupling between the medium and the particle. Taking the message of our work (or of the Mori-Zwanzig approach) to its extreme consequences one could say that observing the dynamics of particles in a medium (i.e., doing microrheology experiment) is generally useless because the inferred parameters do not solely depend on the particle-field interaction. This is the sense in which we consider this feature to be annoying. Concerning the relationship with previous works, we find that the explanation given in Ref.~[8], though providing a nice physical insight, appears system-specific and lacks a simple analytic support (which we clearly provide in our manuscript). The discussion in Ref.~[9] is also very illuminating and inspiring but it is phrased in a context of non-equilibrium conditions and a viscoelastic medium, which might lead the reader think that they are actually important for determining the position-dependence of the effective memory kernel. In this respect, what we present here is a much simpler and natural model of a ``structureless" medium which shows this feature in a way that, as the Referee points out, can be worked out analytically to a large extent.
Motivated by this comment of the Referee we have revised the initial part of the introduction, as highlighted in blue.

Referee's Comment: 3) Just out of curiosity, isn't an UV cut-off on the $q$ modes required? This would also exclude potential Ito/Strato issues at high $q$. (the "$q$ infinity" mode is delta correlated, but perhaps with a negligible weight?).

Reply: This is an interesting question. When an UV cut-off is required, it can be attributed to the finite size of the probe, leading to a size dependence of the various quantities, as discussed in D\'emery and Dean, Phys. Rev. Lett. {\bf 104}, 080601 (2010) and Eur. Phys. J. E {\bf 32}, 377 (2010), Sec. 2.4. The presence of an UV divergence depends on the properties of the field and the dimension of space. Regarding the algebraic decay of the correlations (see Eq. (51)), which is one of our main results, we can see from Eqs.~(49) and (50) and the discussion below that there is no UV divergence.

Referee's Comment: 4) Again, just a comment that perhaps invites some rephrasing here or there: the way the existence of an FDT is described almost makes the reader think this is a nontrivial feature (3.3 ``Remarkably"...I don't think so). In general, integrating out degrees of freedom only reduces the entropy production. When it is zero to begin with (as considered by the authors) there is no reason that integrating the field degrees of freedom out could lead to a violation of the FDT. It is true, however, that by choosing not to work within a path-integral formalism (Onsager Machlup or else), time-reversal for the $X$ dynamics is somewhat obscured.

Reply: We thank the Referee for pointing out this potential misunderstanding. Clearly, as the Referee states, FDT is expected to hold because the system we consider is in thermal equilibrium and this should be certainly reflected in the dynamics. There is nothing remarkable in that. However, it is not obvious that the FDT eventually takes the simple form in Eq. (27) involving the non-linear memory kernel in a way which nicely generalizes the textbook version valid for colored (but linear) noise. This simple form is what we found remarkable. In order to avoid misunderstandings, we have revised part of Sec. 3.3, as highlighted in blue.

Referee's Comment: 5) Sections 4 & 5 are the core of the manuscript, and in themselves they are interesting and self-contained. The medium/particle coupling is used as a perturbation parameter. Surely the linear part could be trimmed down and the "critical" limit could be at least motivated by experiments (if any, would the colloids in lutidine studied by G. Volpe, without light activation part, be a potential candidate). We know that the power laws given by the Gaussian approximation are mean-field values. This could be stated.

Reply: In order to clarify this point we have added a sentence after Eq. (53) and in the very last paragraph of the conclusion, as highligthed in blue.

Referee's Comment: 6) In the conclusion the authors write are confirmed by numerical simulations". I like better the wording adopted in the introduction:the validity of the perturbation expansion is probed by numerical simulations". Here it would be nice if a discussion of physical orders of magnitude (from [8,9]) were inserted. This is a conclusion, so it doesn't need to be up to three digits, but a rough idea of where experiments stand wrt to the perturbation expansion of the authors would be welcome.

Reply: We have revised the wording of the conclusions according to the suggestion of the Referee. Concerning the comparison with actual experiments, the situation is significantly more complicated than in Refs.[8,9] mentioned by the Referee and this is the reason why we did not discuss this issue in the manuscript. In fact, as far as we know, no direct experimental investigation of the phenomena we discussed in the manuscript are currently available. The range of values of the particle radii, correlation length $\xi$, trap stiffness $\kappa$, drag coefficient $\gamma_0$ etc. which are within experimental reach in a realization involving critical fluids are known; however, there is no direct access to the most crucial parameter, i.e., the interaction strength $\lambda V(z)$ characterizing the coupling between the particle and the field and there is also no obvious way a priori to estimate it. This is due to the fact that the field $\phi(x,t)$ corresponds to the order parameter of a second-order phase transition, the actual nature of which depends on the specific system under consideration. How this field is practically defined, in turn, influences the very same quantitative magnitude of the coupling $\lambda V(z)$, see Eq. (1). In order to access it, for example, one could study the case in which the particle is fixed at a certain position (which can be achieved, e.g., by using a strongly confining potential) and determine the resulting order parameter profile which is affected by the presence of the particle. From Eq. (1) of the manuscript, one readily finds that the average order parameter is given by $\langle \phi_q\rangle = \lambda V_q/(q^2+r)$, which, in principle, allows one to readily infer the potential as $\lambda V_q = (q^2+r)\langle \phi_q\rangle$. However, the experimentally accessible field is generically proportional to $\phi$, and therefore one should focus on a quantity which is independent of such proportionality constant, for example by normalizing $\langle \phi_q\rangle$ by the mean square fluctuations of the field in the absence of the particle (i.e., with $\lambda=0$), which is in principle also experimentally accessible. Alternatively, one can experimentally measure (normalized) quantities related to $\langle \phi_q\rangle$, such as the so-called critical adsorption, essentially corresponding to $\langle \phi_{q=0}\rangle$. However, interpreting the (few) available experimental data on the critical adsorption of colloidal particles in critical media it is not straightforward because their dependence on the distance $r$ from criticality does not match the one emerging from the previous expression, due to the difference in the critical exponents which characterize actual binary mixtures (belonging to the Ising universality class) and the Gaussian model considered here.

---

## Round 1 · List of Changes

1. Second paragraph of the Introduction has been substantially modified.
  2. A few sentences have been added at the end of the fourth paragraph of the Introduction.
  3. A detailed discussion about the choice of the coupling potential has been added after Eq. (1).
  4. First sentence of Sec 3.3 has been modified.
  5. The issue of non-linearity in the equations of motion is clarified in the first sentence of Sec. 4.1.
  6. A brief discussion about the dynamical exponents is added after Eq.~(53).
  7. Fifth paragraph of the Conclusions (Sec. 6) has been modified.
  8. A sentence has been added in the last paragraph of Sec. 6.
  9. Ref. [30] has been updated.

---

## Editorial Decision

published